# Optimal Robust Subsidy Policies for Irrational Agent in Principal-Agent MDPs

**Bowen Hu[1], Yixin Tao[2]***

[1] The Engineering& Technical College of Chengdu University of Technology
`hubowen@mail.ustc.edu.cn`
[2] Key Laboratory of Interdisciplinary Research of Computation and Economics,
 Shanghai University of Finance and Economics
`taoyixin@mail.shufe.edu.cn`

## Abstract

We study a principal-agent problem in a Markov Decision Process where the principal provides subsidies to influence the agent's policy, which in turn determines the accrued rewards. Our focus is on designing a robust subsidy scheme that maximizes the principal's cumulative expected return, even when the agent displays bounded rationality and may deviate from the optimal action policy after receiving subsidies.

As a baseline, we first analyze the case of a perfectly rational agent and show that the principal's optimal subsidy coincides with the policy that maximizes social welfare, the sum of the utilities of both the principal and the agent. We then introduce a bounded-rationality model: the globally $\epsilon$-incentive-compatible agent, who accepts any policy whose expected cumulative utility lies within $\epsilon$ of the personal optimum. In this setting, we prove that the optimal robust subsidy scheme problem simplifies to a one-dimensional concave optimization, revealing that optimal subsidies concentrate along social-welfare-maximizing trajectories. We also bound the associated loss in social welfare. Finally, we investigate a finer-grained, state-wise $\epsilon$-incentive-compatible model. In this setting, we show that under two natural definitions of state-wise incentive-compatibility, the problem becomes intractable: one definition results in a non-Markovian agent action policy, while the other renders the search for an optimal subsidy scheme NP-hard.

## 1 Introduction

The principal–agent problem (often modeled as a Stackelberg game) has long been central to the study of strategic interactions where one party acts on behalf of another, yet with potentially misaligned incentives. This setting arises frequently in economics and governance: for example, governments design tax credits, subsidies, and public investments to guide individual behavior toward socially beneficial outcomes, even though market participants ultimately maximize their own private utility. A similar dynamic appears in machine learning, where reinforcement learning with human feedback (RLHF) is employed to align large language models (LLMs) with societal values such as ethics and legal compliance. In both cases, the principal faces the fundamental challenge of shaping an agent's behavior without direct control, while respecting both parties' interests.

In this paper, we investigate the principal–agent problem within the framework of a Markov Decision Process (MDP), where the principal can provide subsidies to influence the agent's action choices. More specifically, in our setting, each action under each state yields two distinct rewards: one for the principal and one for the agent. The principal may also assign non-negative subsidies to actions. The agent selects an action policy based on its own reward combined with subsidies offered by the principal. The principal, in turn, strategically designs these subsidies to influence the agent's choices, aiming to maximize the principal's overall payoff, which equals the total principal's reward associated with the agent's chosen action minus the subsidies provided.

---

*Corresponding author.

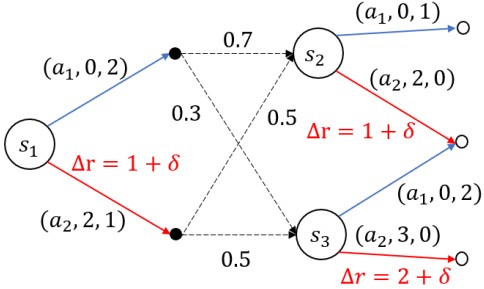

Figure 1: Example of the principal-agent problem. Solid arrows represent actions, annotated with triplets denoting the action index, principal reward, and agent reward, respectively. For example, $(a_2, 2, 1)$ indicates that the agent taking action $a_2$ yields a reward of 2 for the principal and 1 for the agent. Under a subsidy of $1 + \delta$, the agent's reward for selecting $a_2$ increases to $2 + \delta$, while the principal's reward decreases to $1 - \delta$.

We provide a simple illustrative example of such an MDP with three states, where two distinct actions are available in each state, in Figure 1. A natural assumption in such models is that the agent always behaves rationally, selecting the policy that maximizes the sum of the agent's own reward and the subsidies provided by the principal. Thus, in the absence of subsidies, the agent would consistently select action $a_1$ (highlighted in blue) across all states, yielding an expected agent value of 3.3, but resulting in a value of 0 for the principal. To improve this outcome, the principal introduces a subsidy scheme $\Delta r$ targeting the red-marked actions with an infinitesimal amount $\delta > 0$. Consequently, the agent is incentivized to select $a_2$ in all states, yielding an expected agent value of $3.3 + 2\delta$. Meanwhile, such subsidy increases the principal's expected utility to $2 - 2\delta$.

However, the validity of the above example hinges entirely on the agent's strict adherence to optimal behavior. In practice, the rationality assumption of agent is often violated: agents may deviate from perfect rationality due to bounded cognition, incomplete information, or limited computational power. For example, in economics, individuals may fail to optimize utility precisely because of uncertainty or behavioral biases. Similarly, in reinforcement learning, approximate training algorithms may yield suboptimal policies due to limited exploration or finite computation.

Motivated by these considerations, we ask:

> *How should the principal design subsidies when the agent may behave irrationally?*

Our goal is to identify a **robust subsidy scheme** that guarantees the principal the best possible expected cumulative return in the worst-case scenario.

**Our Contributions** We introduce a theoretical framework based on Markov Decision Processes (MDPs) to model the principal-agent problem and formulate the design of an optimal robust subsidy scheme as a minimax optimization problem. Within this framework, we systematically analyze three agent models: the perfectly rational agent, the globally $\epsilon$-incentive-compatible (IC) agent, and the state-wise $\epsilon$-IC agent. For each model, we provide structural insights and algorithmic solutions.

We first study a *perfectly rational agent* as a baseline, who always selects actions that maximize its own utility. In Theorem 3.1, we characterize the optimal subsidy scheme and show in Proposition 3.2 that it suffices to subsidize only actions that maximize social welfare, defined as the sum of the principal's and agent's utilities. Under this scheme, the agent's best-response policy aligns with the social welfare-maximizing policy, establishing a clear benchmark for incentive alignment.

Next, we consider *globally $\epsilon$-IC agents*, who tolerate at most an $\epsilon$ loss relative to their optimal reward under a given subsidy scheme. Unlike perfectly rational agents, these agents may adopt stochastic policies, making the principal's optimization a nontrivial bi-level problem. Theorem 4.1 shows that this problem can be equivalently reduced to maximizing a one-dimensional concave function over a bounded interval, allowing efficient solution via standard first-order methods. Structurally, in Proposition 4.2, we show the optimal subsidy mirrors the perfectly rational case by exclusively rewarding actions that align with maximizing social welfare; and, in the worst-case response, the

agent's policy will assign positive probability to the socially optimal actions, though it may also mix with other actions. We further provide a quantitative analysis of the gap between the total payoff achieved under this robust scheme and the maximum possible social welfare, as shown in Proposition 4.3.

Finally, in Section 5, we examine *state-wise $\epsilon$-IC agents*, for which the $\epsilon$-tolerance must hold at each individual state. Two natural formalizations arise, each presenting distinct challenges. In the first formalization, the agent's worst-case response may necessitate a non-Markovian policy, thereby violating the foundational assumptions of the MDP framework and introducing history dependence that makes the problem computationally intractable. In the second formalization, while the agent's worst-case response remains polynomial-time computable, Theorem 5.1 demonstrates that the principal's problem becomes NP-hard. These findings illustrate that, although state-wise constraints are conceptually appealing, they introduce significant computational and modeling complexities that limit practical applicability.

**Related work** The principal–agent problem, a central concept in economics (Ross, 1973; Grossman & Hart, 1992), arises when a principal delegates tasks to an agent whose actions may be guided by self-interest. This framework underpins both contract theory (Laffont & Maskin, 1981; Guruganesh et al., 2021) and mechanism design (Myerson, 1982; Kadan et al., 2017).

Recent work has examined this problem in the setting of Markov Decision Processes (MDPs). Research in this area falls into two broad directions. The first, information design, seeks to influence the agent's beliefs, as in Bayesian persuasion (Gan et al., 2022a; 2023; Wu et al., 2022; Bernasconi et al., 2023). The second, more closely aligned with our work, focuses on shaping the agent's incentives through policy teaching (Zhang & Parkes, 2008; Banihashem et al., 2022) or environment/model design (Thoma et al., 2024; Yu & Ho, 2022). A comprehensive survey is provided by Dütting et al. (2024).

In this paper, we specifically focus on how the principal uses extra payments to guide the agent's behavior. A related literature is *steering agents in games by payments*, which studies how a principal uses subsidies to induce outcomes in multi-player games. Monderer & Tennenholtz (2003) introduced k-implementation to realize a desired outcome set by committing to non-negative transfers. More recently, focusing on the steering problem for no-regret learners, Zhang et al. (2023) showed how a mediator can use payments to guide agents toward a desired equilibrium. Building on this, Zhang et al. (2025) then introduced a method where a principal uses payments to learn agent utility functions from repeated play, thus providing a path to steering even without prior knowledge of the utilities.

Among these, two approaches are most closely related to our study:

*Contract-based models.* This line of research integrates contract theory with MDPs, assuming the principal observes only states and offers state-dependent payments. Prior studies analyze subgame perfect equilibrium (Wu et al., 2024; Ivanov et al., 2024), showing that history-dependent contracts are necessary for farsighted agents (Bollini et al., 2024). These works typically assume perfectly rational agents and establish that the optimal contract design problem is NP-hard.

*Reward design.* In Reward shaping, the principal modifies the agent's incentives via additional rewards for specific state–action pairs. The principal's goal is to minimize the final cost (Gan et al., 2022b) or subject to a fixed budget (Ben-Porat et al., 2024), with the design problem remaining NP-hard. Extensions address behavioral uncertainty through robust reward design (Wu et al., 2025). In contrast, we incorporate incentive costs directly into the principal's objective, treating them as part of payoff optimization rather than an external constraint.

## 2 PROBLEM FORMULATION

**The Principal-Agent MDP Model** We consider a principal-agent problem modeled as a time-inhomogeneous[1], finite-horizon Markov Decision Process (MDP). In this setting, the principal aims

---

[1] For clarity, time-inhomogeneous implies that the MDP's transitions and policies depend on the current state and timestep. This is distinct from non-Markovian (or history-dependent) settings, where dynamics depend on the sequence of past states and actions.

to achieve a goal by influencing an agent's actions. The principal can offer subsidies to incentivize the agent to follow a policy that benefits the principal.

Formally, we define the problem instance using the tuple $\mathcal{M} = \langle \mathcal{S}, \mathcal{A}, \mathcal{H}, P, r_P, r_A, \hat{s} \rangle$, where:

- $\mathcal{S}$ is the set of the finite states and $\mathcal{A}$ is the set of actions. We assume that both states and actions are *discrete*.
- $\mathcal{H} = \{0, 1, \cdots, H-1\}$ is the set of time steps, with $H$ representing the time horizon.
- $P : \mathcal{S} \times \mathcal{A} \times \mathcal{H} \to \Delta(\mathcal{S})$ is the transition kernel , where $P(s'|s, a, h)$ indicates the probability of transferring to state $s' \in \mathcal{S}$ after executing action $a \in \mathcal{A}$ in state $s \in \mathcal{S}$ at timestep $h \in \mathcal{H}$ .
- $r_P, r_A : \mathcal{S} \times \mathcal{A} \times \mathcal{H} \to \mathbb{R}$ are the reward functions of the principal and the agent, respectively, where $r_P(s, a, h)$ (resp. $r_A(s, a, h)$) denotes the reward obtained by the principal (resp. agent) when the agent executes action $a \in \mathcal{A}$ in state $s \in \mathcal{S}$ at timestep $h \in \mathcal{H}$.
- Without loss of generality, $\hat{s}$ is the fixed starting state for the agent.

**Subsidy Scheme and Action Policy**  A process is called **Markovian** if it depends solely on its current state, independent of its past trajectory. Conversely, a process is **non-Markovian** if it can depend on historical states, i.e., it possesses "memory". While we consider both Markovian and non-Markovian policies, we begin by establishing the definitions within the Markovian framework. The principal commits to a **subsidy scheme** $\Delta r : \mathcal{S} \times \mathcal{A} \times \mathcal{H} \to \mathbb{R}_{\geq 0}$. Here, $\Delta r(s, a, h)$ is a non-negative payment from the principal to the agent for taking action $a$ in state $s$ at timestep $h$. We denote the set of all feasible subsidy policies as $\mathcal{R}_\Delta$.

Given a subsidy $\Delta r$ on action $a$ in state $s$ at timestep $h$, the effective rewards for the principal and agent become $r_P^{\Delta r} = r_P - \Delta r$ and $r_A^{\Delta r} = r_A + \Delta r$. Once the agent observes the subsidy scheme, it chooses a Markovian **action policy** $\pi : \mathcal{S} \times \mathcal{H} \to \Delta(\mathcal{A})$. Based on the agent's (ir)rationality, for any given $\Delta r$, the agent will choose a policy from a specific set of feasible policies, which is denoted by $\Pi(\Delta r)$.

**Value Functions**  To avoid redundancy, we first define the value functions for a generic reward signal $u : \mathcal{S} \times \mathcal{A} \times \mathcal{H} \to \mathbb{R}$. Then, the *Bellman expectation operator* $\mathcal{T}_u$ maps a next-step value function $V_{h+1}$ to the current action-value function:

$$(\mathcal{T}_u V)(s, a, h) \triangleq u(s, a, h) + \sum_{s' \in \mathcal{S}} P(s'|s, a, h) V(s', h+1).$$

Based on this, we define the generic state-value $V^{\pi, u}$ and action-value $Q^{\pi, u}$ backward from $h = H - 1$ to 0, with $V^{\pi, u}(s, H) = 0$:

$$Q^{\pi, u}(s, a, h) = (\mathcal{T}_u V^{\pi, u})(s, a, h), \ where \ V^{\pi, u}(s, h) = \mathbb{E}_{a \sim \pi(\cdot|s, h)}[Q^{\pi, u}(s, a, h)].$$

Similarly, we define the *Bellman optimality operator* and the corresponding optimal values $V^{*, u}$ and $Q^{*, u}$ by replacing the expectation over $\pi$ with a maximization:

$$Q^{*, u}(s, a, h) = (\mathcal{T}_u V^{*, u})(s, a, h), \ where \ V^{*, u}(s, h) = \max_{a \in \mathcal{A}} Q^{*, u}(s, a, h).$$

Then, for any player $i \in \{P, A\}$ under a subsidy scheme $\Delta r$, the standard value functions for player $i$ are simply instances of the generic definition with $u = r_i^{\Delta r}$, denoted as $V_i^{\pi, \Delta r}$ and $Q_i^{\pi, \Delta r}$. The agent's optimal value functions given subsidy scheme $\Delta r$, denoted as $\overline{V}_A^{\Delta r}$ and $\overline{Q}_A^{\Delta r}$, correspond to $V^{*, r_A^{\Delta r}}$ and $Q^{*, r_A^{\Delta r}}$.

**Social Welfare**  We define social welfare as the aggregate reward of both the principal and the agent: $r_{\text{sw}} \triangleq r_P + r_A$, which is invariant to the subsidy scheme $\Delta r$. Given the action policy $\pi$, the social welfare value functions $V_{\text{sw}}^\pi$ and $Q_{\text{sw}}^\pi$ correspond to the generic values with $u = r_{\text{sw}}$. Similarly, the optimal social welfare values $V_{\text{sw}}^*$ and $Q_{\text{sw}}^*$ correspond to the optimal generic values with $u = r_{\text{sw}}$. An action $a$ is **social-welfare-maximizing** if it is greedy with respect to $Q_{\text{sw}}^*$, i.e., $a \in \arg\max_{a' \in \mathcal{A}} Q_{\text{sw}}^*(s, a', h)$.

**Optimization Objective** We consider a robust formulation where the principal seeks a subsidy scheme that performs best against the agent's worst-case response. The agent's adversarial action policy to a subsidy $\Delta r$ is an agent policy $\pi_{\Delta r}$ that minimizes the principal's expected return within the feasible set $\Pi(\Delta r)$ which models agent's rationality:

$$\pi_{\Delta r} \in \arg\min_{\pi \in \Pi(\Delta r)} V_P^{\pi, \Delta r}(\hat{s}, h = 0)$$

The principal's objective is to find the optimal subsidy scheme $\Delta r^*$ that maximizes this worst-case outcome. The optimal value for the principal is therefore:

$$\text{OPT} \triangleq \max_{\Delta r \in \mathcal{R}_\Delta} \min_{\pi \in \Pi(\Delta r)} V_P^{\pi, \Delta r}(\hat{s}, h = 0) \tag{2.1}$$

**Non-Markovian setting** In non-Markovian setting, both principal and agent can make decision based on the whole history rather than the current state. Formally, a trajectory is is a sequence of states and actions of length $T$, denoted as $\tau = (s_1, a_1, s_2, a_2, \cdots, s_{T-1}, a_{T-1}, s_T)$. Let $\mathcal{T}$ be the set of all trajectories. Unless otherwise specified, we assume that all trajectories originate from the fixed initial state $\hat{s}$. Building on this, the non-Markovian subsidy scheme and action policy are defined by substituting the state and timestep dependency with the trajectory, i.e., $\Delta r : \mathcal{T} \times \mathcal{A} \to \mathbb{R}_{\geq 0}$ and $\pi : \mathcal{T} \to \Delta(\mathcal{A})$. In addition, under non-Markovian policies, the corresponding value function is also history-dependent, e.g., $V_i^{\Delta r}(\tau)$ and $Q_i^{\Delta r}(\tau)$. We note that though we mainly analyze in the Markovian setting, all positive results can be extended to non-Markovian setting.

## 3 WARM-UP: THE PERFECTLY RATIONAL AGENT

We begin with the simplest setting of a perfectly rational agent, defined as an agent that seeks to maximize its cumulative reward. Although this scenario is conceptually straightforward, it provides a crucial foundation for the subsequent analysis of more complex, irrational agents. We formalize this concept as follows.

**Definition 3.1** (Perfectly Rational Agent). *Given a subsidy scheme $\Delta r$, the action policy $\pi \in \Pi_0(\Delta r)$ of a perfectly rational agent satisfies the constraint*

$$V_A^{\pi, \Delta r}(\hat{s}, h = 0) \geq \overline{V}_A^{\Delta r}(\hat{s}, h = 0).$$

**Tie-breaking Rule** A tie-breaking rule dictates the agent's choice when multiple actions yield identical rewards. In this setting with a perfectly rational agent, we assume that when two options provide the same personal reward, the agent selects the more cooperative action, that is, the one that benefits the principal more. For example, consider a single state with two actions. Both give the agent a reward of $0$, but the principal receives $2$ for the first action and $0$ for the second. Even a negligible subsidy on the first action makes it strictly preferred. As the subsidy approaches zero, the agent's choice remains the action with a higher principal value. Thus, tie-breaking systematically favors actions that increase the principal's payoff. This assumption allows for a tractable proof of optimality in this section, but it is important to note that we will not rely on this rule in the more general frameworks developed later in the paper.

### 3.1 OPTIMAL SUBSIDY SCHEME

With the definition of perfect rationality, we now address the problem of determining the optimal subsidy scheme $\Delta r^*$. The following theorem characterizes the principal's optimal payoff and the optimal subsidy scheme. Detailed proof is deferred to Appendix A.3.

**Theorem 3.1** (Optimal Subsidy Scheme). *For a perfectly rational agent, the principal's optimal payoff is given by*

$$V_{sw}^*(\hat{s}, h = 0) \ - \ \overline{V}_A^{\Delta r = 0}(\hat{s}, h = 0),$$

*that is, the maximum attainable social welfare (over all action policies) minus the maximum reward the agent can obtain in the absence of subsidies. Furthermore, there exists an optimal subsidy scheme $\Delta r^*$ such that, for every state–action–timestep triple $(s, a, h)$,*

$$\Delta r^*(s, a, h) \ = \ \overline{V}_A^{\Delta r = 0}(s, h) - \overline{Q}_A^{\Delta r = 0}(s, a, h). \tag{3.1}$$

*Proof Sketch.* The principal's optimal payoff is bounded above by $V_{\text{sw}}^*(\hat{s}, h = 0) - \overline{V}_A^{\Delta r=0}(\hat{s}, h = 0)$, since the total value of the principal and agent cannot exceed the maximum possible social welfare, and the agent will not accept less than their stand-alone value without subsidies. This upper bound is achieved under the subsidy scheme $\Delta r^*$ defined in equation (3.1). Under this scheme, the agent's adjusted $Q$-values are equalized across all actions: $Q_A^{\Delta r^*}(s, a, h) = \overline{V}_A^{\Delta r=0}(s, h)$ for all $(s, a, h)$. Thus, the agent is indifferent among all actions. Our provisional tie-breaking rule then ensures the agent selects actions that maximize the principal's reward, allowing the principal's payoff to exactly reach the upper bound. □

Although Theorem 3.1 identifies an optimal subsidy scheme that provides transfers on nearly all actions, the following proposition shows that, to achieve optimal rewards, the principal needs to subsidize only the social-welfare-maximizing actions. The detailed proof is deferred to Appendix A.4.

**Proposition 3.2** (Social Welfare). *There exists an optimal subsidy scheme $\Delta r_{sw}$ that assigns positive transfers exclusively to social-welfare-maximizing actions. Under $\Delta r_{sw}$, the agent implements social-welfare-maximizing agent policy $\pi_{sw}$, allowing the principal to attain the maximum achievable social welfare.*

## 4 OPTIMAL POLICIES FOR GLOBALLY $\epsilon$-IC AGENTS

When an agent is no longer perfectly rational, the optimality of its response ceases to be the sole factor guiding its decisions. To model such bounded rationality, a natural approach is to assume that the agent can tolerate a maximum reward loss of $\epsilon$, in line with the classical notion of $\epsilon$-incentive compatibility (IC). However, since we are dealing with sequential decision-making, several interpretations of $\epsilon$-IC are possible. Here, we focus on the so-called *globally $\epsilon$-IC agent*, which constrains only the cumulative reward loss over the entire decision horizon.

**Definition 4.1.** *An agent is a globally $\epsilon$-IC agent if and only if, given a subsidy scheme $\Delta r$, the action policy $\pi \in \Pi_\epsilon^g(\Delta r)$ satisfies*

$$V_A^{\pi, \Delta r}(\hat{s}, h = 0) \geq \overline{V}_A^{\Delta r}(\hat{s}, h = 0) - \epsilon.$$

### 4.1 OPTIMAL SUBSIDY SCHEME

We now consider the problem of determining the optimal subsidy scheme $\Delta r^*$. Unlike the perfectly rational case, the agent's best-response policy may be stochastic. To handle this, we reformulate the objective (2.1) using occupancy measures. Specifically, let $\mu(s, a, h)$ denote the probability that the agent takes action $a$ in state $s$ at timestep $h$. Replacing the policy $\pi$ with its corresponding occupancy measure $\mu$, the optimization problem becomes

$$\max_{\Delta r \in \mathcal{R}_\Delta} \min_{\mu \in M(\Delta r)} \sum_{s,a,h} \mu(s, a, h)\big(r_P(s, a, h) - \Delta r(s, a, h)\big), \tag{4.1}$$

where $M(\Delta r)$ is the set of occupancy measures satisfying the following constraints:

Initial state:
$$\sum_a \mu(\hat{s}, a, h = 0) = 1, \quad \sum_a \mu(s, a, h = 0) = 0 \quad \forall s \neq \hat{s}, \tag{4.2a}$$

Transition:
$$\sum_a \mu(s, a, h) = \sum_{s',a'} \mu(s', a', h - 1)P(s|s', a', h - 1), \tag{4.2b}$$

Non-negativity:
$$\mu(s, a, h) \geq 0, \tag{4.2c}$$

Global $\epsilon$-IC:
$$\sum_{s,a,h} \mu(s, a, h)\big(r_A(s, a, h) + \Delta r(s, a, h)\big) \geq \overline{V}_A^{\Delta r}(\hat{s}, h = 0) - \epsilon. \tag{4.2d}$$

Directly solving this program is challenging for two main reasons. First, the feasible set of $\mu$ is not fixed but depends on the choice of $\Delta r$, creating a coupling between the inner and outer variables that distinguishes our setting from standard minimax formulations. Second, defining $f(\Delta r) =$

$\min_{\mu \in M(\Delta r)} \sum_{s,a,h} \mu(s,a,h)\big(r_P(s,a,h) - \Delta r(s,a,h)\big)$ shows that $f(\Delta r)$ is not concave in $\Delta r$ (see Appendix A.2.1 for example). Consequently, the outer problem $\max_{\Delta r} f(\Delta r)$ is not a concave maximization , which rules out standard convex optimization methods.

In our main theorem, we show the problem can be reformulated to a one-dimensional concave optimization (Theorem 4.1). The approach leverages the dual of the inner optimization problem and swaps the order of optimization between the subsidy scheme $\Delta r$ and the dual variables $(\alpha, V)$. The optimal subsidy scheme can then be expressed as the difference between the $V$-function and $Q$-function, analogous to the perfectly rational case.

**Theorem 4.1.** *The optimization problem (4.1) is equivalent to maximizing a concave function $F(x)$, formulated as*

$$\max_{x \in [0,1)} F(x) = x V_{sw}^*(\hat{s}, h=0) - V_x^*(\hat{s}, h=0) - \frac{x}{1-x}\epsilon,$$

*where, for each state $s$ and timestep $h$, $V_x^*(s,h) \triangleq \max_\pi \left\{ x V_{sw}^\pi(s,h) - V_P^{\pi, \Delta r=0}(s,h) \right\}$.*

*Furthermore, for an optimal $x^*$, there exists an optimal subsidy scheme $\Delta r^*$ such that*

$$\Delta r^*(s,a,h) = V_{x^*}^*(s,h) - Q_{x^*}^*(s,a,h) \tag{4.3}$$

*where $Q_{x^*}^*(s,a,h) \triangleq x^* r_{sw}(s,a,h) - r_P(s,a,h) + \sum_{s' \in \mathcal{S}} P(s'|s,a,h) V_{x^*}^*(s', h+1)$.*

*Proof.* We begin by considering the inner program over the state-action occupancy measure $\mu$ for a fixed subsidy scheme $\Delta r$. This program is a linear program. By introducing dual variables $\alpha \in \mathbb{R}_+$ for the globally $\epsilon$-IC constraint (4.2d) and $V \in \mathbb{R}^{|S|(H+1)}$ for the transition (4.2a) and initial state (4.2b) constraints, we can express the problem in its dual form. Combining this with the outer maximization over $\Delta r$, $\alpha$, and $V$ yields the following optimization problem:

$$\max_{\Delta r \in \mathcal{R}_\Delta, \alpha \geq 0, V} V(\hat{s}, h=0) - \alpha\epsilon + \alpha \overline{V}_A^{\Delta r}(\hat{s}, h=0)$$

such that $V(s,h) \leq r_P(s,a,h) - \alpha r_A(s,a,h) - (1+\alpha)\Delta r(s,a,h) + \sum_{s' \in \mathcal{S}} P(s'|s,a,h) V(s', h+1)$ for any $s \in \mathcal{S}$, $a \in \mathcal{A}$, and $h \in \mathcal{H}$; and with the terminal condition $V(s,H) = 0$ for any state $s \in \mathcal{S}$.

Next, we exchange the order of $\max_{\Delta r \in \mathcal{R}_\Delta}$ and $\max_{\alpha \geq 0, V}$ and analyze maximization over $\Delta r$ for a fixed $V$ and $\alpha$. Notice that the objective is non-decreasing with respect to $\Delta r$, since $\overline{V}_A^{\Delta r}(\hat{s}, h=0)$ represents the maximum value attainable by the agent under the subsidy $\Delta r$. Additionally, the constraints impose an upper bound on each $\Delta r(s,a,h)$:

$$\Delta r(s,a,h) \leq \frac{1}{1+\alpha}\Big( -V(s,h) + \sum_{s' \in \mathcal{S}} P(s'|s,a,h) V(s', h+1) + r_P(s,a,h) - \alpha r_A(s,a,h)\Big).$$

Thus, the optimal choice for $\Delta r$ is to take this upper bound, making the inequality hold with equality. Given $\alpha$ and $V$, substituting the optimal choice of $\Delta r$, the RHS of the above inequality, into $\overline{V}_A^{\Delta r}(\hat{s}, h=0) = \max_\pi \mathbb{E}_\pi\Big[ \sum_{t=0}^{H-1} r_A(s_t, a_t, t) + \Delta r(s_t, a_t, t)\Big]$ gives

$$\overline{V}_A^{\Delta r}(\hat{s}, h=0) = \max_\pi \frac{1}{1+\alpha} \mathbb{E}_\pi\Bigg[ \sum_{t=0}^{H-1} \big( r_P(s_t, a_t, t) + r_A(s_t, a_t, t)\big)$$

$$+ \sum_{s_{t+1} \in \mathcal{S}} P(s_{t+1}|s_t, a_t, t) V(s_{t+1}, t+1) - V(s_t, t)\Bigg]$$

$$= \frac{1}{1+\alpha}\Big( V_{sw}^*(\hat{s}, h=0) - V(\hat{s}, h=0)\Big).$$

Substituting this back, the problem reduces to the following optimization problem. Note that the first constraint guarantees the feasibility of $\Delta r$.

$$\max_{\alpha \geq 0} \max_V \frac{1}{1+\alpha} V(\hat{s}, h=0) + \frac{\alpha}{1+\alpha} V_{sw}^*(\hat{s}, h=0) - \alpha\epsilon$$

$$\text{s.t.} \quad V(s,h) \leq r_P(s,a,h) - \alpha r_A(s,a,h) + \sum_{s' \in \mathcal{S}} P(s'|s,a,h) V(s', h+1),$$

$$V(s,H) \leq 0.$$

Observing the inner optimization over $V(s, h)$ coincides with form of minimizing cumulative reward in an MDP with modified reward $r_P - \alpha r_A$. By letting $x = \frac{\alpha}{1+\alpha}$ and introducing $V_x^*(s, h)$ equals $= -\frac{1}{1+\alpha}$ times the optimal value of $V(s, h)$, the formulation equals

$$\max_{x \in [0,1)} \quad x \cdot V_{\text{sw}}^*(\hat{s}, h = 0) - V_x^*(\hat{s}, h = 0) - \frac{x}{1-x}\epsilon$$

where
$$V_x^*(\hat{s}, h = 0) \triangleq -(1-x) \cdot \min_\pi \left\{ V_P^{\pi, \Delta r = 0}(\hat{s}, h = 0) - \frac{x}{1-x} V_A^{\pi, \Delta r = 0}(\hat{s}, h = 0) \right\}$$

$$= \max_\pi \left\{ x V_{\text{sw}}^\pi(\hat{s}, h = 0) - V_P^{\pi, \Delta r = 0}(\hat{s}, h = 0) \right\}.$$

Restricting $\pi$ to deterministic action policies does not change the value of $V_x^*(\hat{s}, h = 0)$, and under this restriction, $V_x^*(\hat{s}, h = 0)$ is the maximum of finitely many linear functions in $x$, so the objective function is concave over the interval $[0, 1)$. $\qquad \square$

**Markovian vs. Non-Markovian** In our framework, both the principal and the agent may adopt non-Markovian strategies. For example, the principal might determine subsidies based not only on the agent's current action but also on past actions. Similarly, in equation (4.1), the agent could adopt a non-Markovian globally $\epsilon$-IC policy to reduce the principal's reward. Nevertheless, the following two key observations establish that it suffices to restrict attention to Markovian strategies.

*First observation*: Given a Markovian subsidy scheme of the principal, there always exists a Markovian globally $\epsilon$-IC policy for the agent that minimizes the principal's reward. This follows from the fact that the inner optimization problem in equation (4.1) is a linear program. Any non-Markovian $\epsilon$-IC policy can be represented by an occupancy measure $\mu(s, a, h)$, which specifies the probability of taking action $a$ in state $s$ at timestep $h$. Such an occupancy measure can always be replicated by a Markovian policy, ensuring identical rewards for both the principal and the agent.

*Second observation*: Among all possible subsidy schemes—Markovian or non-Markovian—the Markovian scheme specified in equation (4.3) is optimal. A non-Markovian scheme can be transformed into a Markovian one by augmenting the state space to encode the relevant history. By Theorem 4.1, for each state–action pair in this augmented representation, the scheme in equation (4.3) coincides exactly with its Markovian counterpart.

**Remark** We briefly examine the boundary cases of $x^*$ and $\epsilon$ in Theorem 4.1. When $\epsilon = 0$, as $x^* \to 1$, the principal's value approaches $V_{\text{sw}}^*(\hat{s}, h = 0) - V_A^{\Delta r = 0}(\hat{s}, h = 0)$, consistent with the tie-breaking rule in the perfectly rational case. This shows that the globally $\epsilon$-IC agent naturally generalizes the perfectly rational agent.

### 4.2 ACTION POLICY

According to Theorem 4.1, the optimal subsidy scheme $\Delta r^*$ takes a form similar to that in the perfectly rational case. The following proposition shows that the principal can still allocate positive transfers exclusively to the social-welfare-maximizing actions. Furthermore, the agent is still willing to cooperate with the principal to a certain extent by choosing one social-welfare-maximizing agent policy $\pi_{\text{sw}}$ with probability $x^*$, the optimal solution in Theorem 4.1. The detailed proof of the following proposition is deferred to Appendix A.5.

**Proposition 4.2** (Optimal subsidy scheme and action policy)**.** *There exists an optimal subsidy scheme $\Delta r_{sw}$ that assigns positive reward transfers solely to social-welfare-maximizing actions. Meanwhile, there exists a globally $\epsilon$-IC action policy $\pi_{\Delta r_{sw}}$ minimizing the principal's reward, which is the mixture of a social-welfare-maximizing agent policy $\pi_{sw}$ and one other action policy, placing a weight of at least $x^*$ on $\pi_{sw}$.*

*Proof Sketch.* The proof relies on two key insights. First, under the optimal subsidy scheme $\Delta r^*$, the policy $\pi_{\text{sw}}$ achieves the maximum agent expected cumulative reward, $\overline{V}_A^{\Delta r^*}(\hat{s}, h = 0)$. This implies that it is sufficient to provide subsidies only along the trajectories induced by $\pi_{\text{sw}}$, without affecting the optimal value for the principal. Second, there exists an action policy $\hat{\pi}$ whose agent value falls below $\overline{V}_A^{\Delta r^*}(\hat{s}, h = 0) - \epsilon$, which can be combined with $\pi_{\text{sw}}$ to form the globally $\epsilon$-IC policy $\pi_{\Delta r_{\text{sw}}}$, such that the dual of the global $\epsilon$-incentive compatibility constraint is tight. $\qquad \square$

### 4.3 SOCIAL WELFARE

We define the social welfare gap $\delta_{\text{sw}}$ as the difference between the maximum attainable welfare and the welfare achieved under the optimal subsidy scheme $\Delta r^*$. When $\epsilon \to +\infty$, the agent can effectively bypass the global $\epsilon$-IC constraint and freely select any action policy. In this limit, the welfare gap becomes $\delta_{\text{sw}} = V_{\text{sw}}^*(\hat{s}, h = 0) - \min_\pi V_{\text{sw}}^\pi(\hat{s}, h = 0)$. Our objective is to characterize the upper bound on $\delta_{\text{sw}}$ and the rate at which social welfare declines as a function of $\epsilon$, particularly in the regime where $\epsilon$ remains small. We first establish the following upper bound on $\delta_{\text{sw}}$.

**Proposition 4.3.** *Given $\epsilon$ and the corresponding optimal solution $x^* \in (0,1)$, the social welfare gap is $\delta_{sw} = \frac{\epsilon}{1-x^*}$ and it is upper bounded by $O(\sqrt{\epsilon})$.*

This $O(\sqrt{\epsilon})$ bound can be achieved in certain specific cases (see Appendix A.2.2 for an example). However, in most cases, the social welfare gap $\delta_{\text{sw}}$ exhibits two different growth rates—$O(\sqrt{\epsilon})$ or $O(\epsilon)$—depending on whether $V_x^*$ is differentiable at $x^*$. A concrete example is provided below, while detailed discussions are deferred to Appendix A.6.1.

**Example** Consider a single-period scenario with three actions. For the first action, the principal's reward is 7 and the agent's reward is 3. For the second action, the principal's reward is 1 and the agent's reward is 2. For the third action, the principal's reward is 1 and the agent's reward is 0. Figure 2(a)shows that $\delta_{\text{sw}}$ can grow at rates of $O(\epsilon)$ and $O(\sqrt{\epsilon})$, corresponding to the cases in Figure 2(b) where $x$ remains constant and decreases at $O(\sqrt{\epsilon})$. Figure 2(c) depicts the piecewise-linear relationship between $V_x^*(\hat{s}, h = 0)$ and $x$, where the constant-$x$ value in Figure 2(b) coincides with the break point of $V_x^*(\hat{s}, h = 0)$, a non-differentiable point of the objective function.

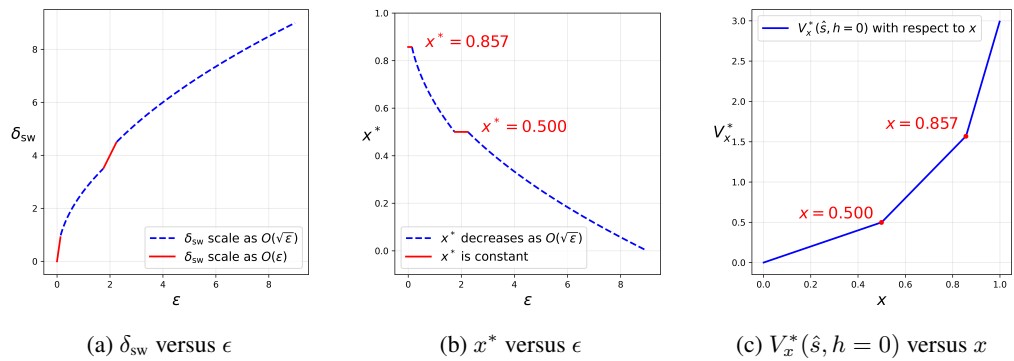

(a) $\delta_{\text{sw}}$ versus $\epsilon$      (b) $x^*$ versus $\epsilon$      (c) $V_x^*(\hat{s}, h = 0)$ versus $x$

Figure 2: Curves of $\delta_{\text{sw}}$ and $x^*$ versus $\epsilon$ when $V_x^*(\hat{s}, h = 0)$ is non-differentiable.

## 5 STATE-WISE $\epsilon$-IC AGENT

In this section, we examine the *state-wise $\epsilon$-IC agent*, which differs from the globally $\epsilon$-IC agent in that incentive compatibility is enforced locally at each state and decision step. Intuitively, such an agent ensures that its chosen action remains within $\epsilon$ of the best immediate value available at that decision point. While the idea is straightforward, constructing a mathematically consistent and tractable formalization is more subtle. We provide two definitions below.

**Value-Consistent State-Wise $\epsilon$-IC Agent** We first define the *value-consistent state-wise $\epsilon$-IC agent*, where the agent's action at each state must approximate the optimal reward within $\epsilon$.

**Definition 5.1.** *An agent is a value-consistent state-wise $\epsilon$-IC agent if, under a subsidy scheme $\Delta r$, the induced policy $\pi \in \Pi_\epsilon^v(\Delta r)$ satisfies $V_A^{\pi, \Delta r}(s, h) \geq \overline{V}_A^{\Delta r}(s, h) - \epsilon$ for all $s \in \mathcal{S}$ and $h \in \mathcal{H}$.*

A key challenge with this formulation is that the agent's policy minimizing the principal's reward under a given subsidy scheme may be **non-Markovian**. Specifically, consider the agent can apply a non-Markovian policy $\pi$ such that for any trajectory $\tau$, $V_A^{\pi, \Delta r}(\tau) \geq \overline{V}_A^{\Delta r}(\tau) - \epsilon$. In such cases, the agent's policy cannot be represented within polynomial size. Meanwhile, from the perspective of the principal, the principal can gain extra value by applying the non-Markovian subsidy scheme once principal further assume the agent is non-Markovian.

To illustrate, consider the post-subsidy MDP in Figure 3 with $\epsilon = 1$. The rewards are: (i) action $a_1$ at $s_1$ yields 3.21 for the principal and 2 for the agent; (ii) $a_2$ at $s_3$ yields 0.21 and 1, respectively; and (iii) 0 otherwise. Under a **Markovian policy**, the agent steers from $s_1$ to $s_3$ and selects $a_2$, limiting the principal's reward to 0.21. However, under a **non-Markovian policy**, we can split $s_3$ into history-dependent states $s_3^1$ and $s_3^2$. At $s_3^1$, the agent always chooses $a_2$, while at $s_3^2$, it deterministically chooses the action other than $a_2$. This drops the principal's expected reward to 0.105.

We further demonstrate that principal can gain more by applying the non-Markovian subsidy scheme: Consider the standard model where the principal applies a Markovian subsidy scheme to a Markovian agent. Calculations show the optimal subsidy targets only action $a_1$ with value 1, resulting in a principal value of 0.71. If the agent becomes non-Markovian but the subsidy remains unchanged, the principal's value falls to 0.605. However, by implementing a non-Markovian scheme that includes an additional 0.1 subsidy for action $a_2$ in state $s_3^2$, the principal can recover a higher optimal value of 0.615. Detailed calculations are deferred to Appendix A.7.2.

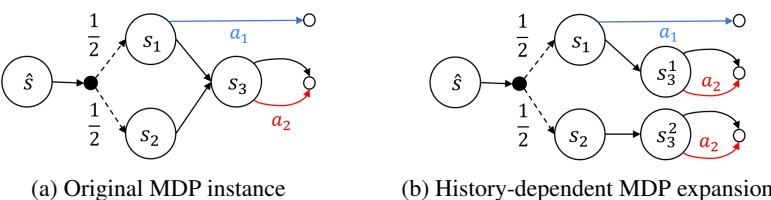

(a) Original MDP instance          (b) History-dependent MDP expansion

Figure 3: Illustration of value-consistent state-wise $\epsilon$-IC agents.

**Greedy State-Wise $\epsilon$-IC Agent**  To avoid non-Markovian behavior, we introduce the *greedy state-wise $\epsilon$-IC agent*, which replaces recursive value computations with greedy look-ahead. Once the subsidy scheme is fixed, $\overline{V}_A^{\Delta r}$ becomes deterministic, and the agent greedily minimizes the principal's value through local decisions.

**Definition 5.2.** *An agent is a greedy state-wise $\epsilon$-IC agent if, under subsidy scheme $\Delta r$, the induced policy $\pi \in \Pi_\epsilon^s(\Delta r)$ satisfies, for all $s \in S$, $h \in \mathcal{H}$:*

$$\sum_{a \in \mathcal{A}} \pi(a|s,h)\left(r_A^{\Delta r}(s,a,h) + \sum_{s' \in \mathcal{S}} P(s'|s,a,h)\overline{V}_A^{\Delta r}(s',h+1)\right) \geq \overline{V}_A^{\Delta r}(s,h) - \epsilon.$$

However, even in this simplified greedy setting, designing the principal's optimal subsidy scheme remains computationally intractable. The complete proof is deferred to Appendix A.7.3.

**Theorem 5.1.** *Given a greedy state-wise $\epsilon$-IC agent, computing the principal's optimal subsidy scheme is NP-hard.*

## 6  CONCLUSION

In this paper, we study a principal-agent problem with the aim of designing a robust subsidy scheme that maximizes the cumulative expected return in the presence of an irrational agent. We demonstrate that, under the globally $\epsilon$-IC assumption, the optimal subsidy scheme can be effectively determined, representing a natural extension of the perfectly rational case. We further show that formulating the state-wise $\epsilon$-IC follower is computationally challenging. As future work, it would be interesting to consider scenarios in which the principal does not have prior knowledge of the agent's reward function or the value of $\epsilon$, such as in a learning-based setting.

### ACKNOWLEDGMENTS

This work was supported by the National Natural Science Foundation of China under Grant No.62502296.

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

## A  APPENDIX

### A.1  USAGE OF LLM

We employed the large language model (LLM) to assist in refining the language and enhancing the clarity of this manuscript. The LLM was **not** used for generating research ideas, identifying related work, performing analyses, or contributing to the substantive scientific content of this paper.

### A.2  MISSING EXAMPLES

#### A.2.1  COUNTEREXAMPLE ON CONVEXITY

Consider a single-period scenario with three actions and $\epsilon = 2$. The principal receives $0$ for the first action and $5$ for both the second and third actions, while the agent's reward is $0$ for the first action and $1$ for other actions. Let $\Delta r_1$ and $\Delta r_2$ transfer 2 units to the second and third actions, respectively; both yield a principal value of $1$. A convex combination, $\Delta r_3$, transferring 1 unit to both actions, results in a leader value of $0$, directly violating concavity. This illustrates that the outer optimization cannot be assumed concave.

#### A.2.2  EXAMPLE OF $\delta_{\text{sw}}$ SCALING AS $O(\sqrt{\epsilon})$

Consider a single-period scenario with two actions and $\epsilon = 1$. The principal and agent values for the first action are 4 and 0, respectively, and for the second action, they are 0 and 2. In this setting, $1-x^*$ always scales as $O(\sqrt{\epsilon})$ and matches the upper bound. The core idea behind is in such instance, the function $V_x^*(\hat{s}, h = 0)$ is a complete linear function in interval $[0, 1]$. Figure 4 illustrates relationship between $\delta_{\text{sw}}$, $x^*$, and $\epsilon$, along with the behavior of $V_x^*(\hat{s}, h = 0)$ as a function of $x$.

### A.3  PROOF OF THEOREM 3.1

By definition, for any subsidy scheme $\Delta r$ with induced action policy $\pi_{\Delta r} \in \Pi_0(\Delta r)$, we have

$$V_P^{\pi_{\Delta r}, \Delta r}(\hat{s}, h = 0) + V_A^{\pi_{\Delta r}, \Delta r}(\hat{s}, h = 0) = V_{\text{sw}}^{\pi_{\Delta r}}(\hat{s}, h = 0) \leq V_{\text{sw}}^*(\hat{s}, h = 0). \tag{A.1}$$

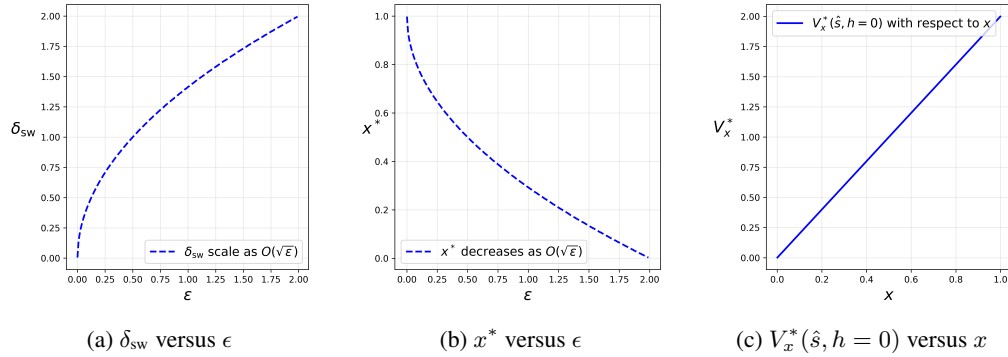

(a) $\delta_{\text{sw}}$ versus $\epsilon$        (b) $x^*$ versus $\epsilon$        (c) $V_x^*(\hat{s}, h = 0)$ versus $x$

Figure 4: Curves of $\delta_{\text{sw}}$ and $x^*$ versus $\epsilon$ when $V_x^*(\hat{s}, h = 0)$ is differentiable.

Moreover, since any subsidy scheme provides the agent with non-negative reward transfers, backward induction gives

$$
\begin{aligned}
V_A^{\pi, \Delta r}(\hat{s}, h = 0) &= \max_{\pi} \mathbb{E}\left[ \sum_{t=0}^{H-1} r_A^{\Delta r}(s_t, a_t, t) \right] \\
&\geq \max_{\pi} \mathbb{E}\left[ \sum_{t=0}^{H-1} r_A(s_t, a_t, t) \right] \\
&= \overline{V}_A^{\Delta r = 0}(\hat{s}, h = 0).
\end{aligned}
\tag{A.2}
$$

Combining this with inequality (A.1), the optimal principal value is upper bounded by

$$
\text{OPT} \leq V_{\text{sw}}^*(\hat{s}, h = 0) - \overline{V}_A^{\Delta r = 0}(\hat{s}, h = 0).
$$

It remains to show that

$$
\Delta r^*(s, a, h) = \overline{V}_A^{\Delta r = 0}(s, h) - \overline{Q}_A^{\Delta r = 0}(s, a, h)
$$

achieves this bound. For any policy $\pi$, substituting $\Delta r^*$ into $V_A^{\pi, \Delta r^*}(\hat{s}, h = 0)$ and applying backward induction establishes that the agent's value for every action equals $V_A^{\Delta r = 0}(\hat{s}, h = 0)$, which makes inequality (A.2) tight. In addition, since the social-welfare-maximizing policy $\pi_{\text{sw}}$ renders inequality (A.1) exact, the principal's value under $\Delta r^*$ is

$$
V_P^{\pi_{\text{sw}}, \Delta r^*}(\hat{s}, h = 0) = V_{\text{sw}}^*(\hat{s}, h = 0) - \overline{V}_A^{\Delta r = 0}(\hat{s}, h = 0),
$$

which coincides with the upper bound. Consequently, under the cooperative tie-breaking rule, the agent selects $\pi_{\text{sw}}$, thereby achieving

$$
\text{OPT} = V_{\text{sw}}^*(\hat{s}, h = 0) - \overline{V}_A^{\Delta r = 0}(\hat{s}, h = 0).
$$

### A.4 PROOF FOR PROPOSITION 3.2

Recall that in Theorem 3.1, we defined the optimal reward transfer as

$$
\Delta r^*(s, a, h) = \overline{V}_A^{\Delta r = 0}(s, h) - \overline{Q}_A^{\Delta r = 0}(s, a, h).
$$

In fact, it suffices to retain the reward transfer only along the social-welfare-maximizing actions. In particular, we define subsidy scheme $\Delta r_{\text{sw}}^*$ as

$$
\Delta r_{\text{sw}}(s, a, h) = \begin{cases} \Delta r^*(s, a, h), & \text{if } \pi_{\text{sw}}(a|s, h) > 0, \\ 0, & \text{otherwise.} \end{cases}
$$

so that the agent value of social-welfare-maximizing action policy $\pi_{\text{sw}}$ under $\Delta r_{\text{sw}}$ is

$$
V_A^{\pi_{\text{sw}}, \Delta r_{\text{sw}}}(\hat{s}, h = 0) = \overline{V}_A^{\Delta r^*}(\hat{s}, h = 0) = \overline{V}_A^{\Delta r = 0}(\hat{s}, h = 0),
$$

which yields a principal value of $V_{\text{sw}}^*(\hat{s}, h = 0) - \overline{V}_A^{\Delta r = 0}(\hat{s}, h = 0)$.

## A.5 Proof of Proposition 4.2

**Optimal subsidy scheme**  Recall from (4.3) that

$$
\begin{aligned}
\Delta r^*(s, a, h) &= V_{x^*}^*(s, h) - Q_{x^*}^*(s, a, h) \\
&= V_{x^*}^*(s, h) - x^* r_{\mathrm{sw}}(s, a, h) + r_P(s, a, h) - \sum_{s' \in \mathcal{S}} P(s'|s, a, h) V_{x^*}^*(s', h+1) \\
&= (1 - x^*) r_P(s, a, h) - x^* r_A(s, a, h) + V_{x^*}^*(s, h) - \sum_{s' \in \mathcal{S}} P(s'|s, a, h) V_{x^*}^*(s', h+1).
\end{aligned}
$$

Let $\pi_{\mathrm{sw}}$ be a deterministic social-welfare-maximizing agent policy. We now define a subsidy scheme $\Delta r_{\mathrm{sw}}$ that is restricted to $\pi_{\mathrm{sw}}$:

$$
\Delta r_{\mathrm{sw}}(s, a, h) \triangleq
\begin{cases}
\Delta r^*(s, a, h), & \text{if } \pi_{\mathrm{sw}}(a|s, h) > 0, \\
0, & \text{otherwise.}
\end{cases}
$$

First, we claim that under either subsidy scheme, $\Delta r_{\mathrm{sw}}$ or $\Delta r^*$, the agent's optimal value is identical:

$$
\overline{V}_A^{\Delta r^*}(\hat{s}, h = 0) = \overline{V}_A^{\Delta r_{\mathrm{sw}}}(\hat{s}, h = 0).
$$

The central argument of the proof is that, under both $\Delta r^*$ and $\Delta r_{\mathrm{sw}}$, action policy $\pi_{\mathrm{sw}}$ can achieve the agent's maximal value, and the values thus attained coincide. Specifically,

$$
\overline{V}_A^{\Delta r^*}(\hat{s}, h = 0) \overset{(a)}{=} V_A^{\pi_{\mathrm{sw}}, \Delta r^*}(\hat{s}, h = 0) \overset{(b)}{=} V_A^{\pi_{\mathrm{sw}}, \Delta r_{\mathrm{sw}}}(\hat{s}, h = 0) \overset{(c)}{=} \overline{V}_A^{\Delta r_{\mathrm{sw}}}(\hat{s}, h = 0).
$$

In what follows, we establish the validity of each equality (a)-(c) sequentially.

We first show (a):

$$
\overline{V}_A^{\Delta r^*}(\hat{s}, h = 0) = V_A^{\pi_{\mathrm{sw}}, \Delta r^*}(\hat{s}, h = 0).
$$

To see this, consider any action policy $\pi$ under $\Delta r^*$,

$$
\begin{aligned}
V_A^{\pi, \Delta r^*}(\hat{s}, h = 0) &= \mathbb{E}_{\pi} \left[ \sum_{t=0}^{H-1} r_A(s_t, a_t, t) + \Delta r^*(s_t, a_t, t) \right] \\
&= \mathbb{E}_{\pi} \left[ \sum_{t=0}^{H-1} r_A(s_t, a_t, t) + (1 - x^*) r_P(s_t, a_t, t) - x^* r_A(s_t, a_t, t) \right. \\
&\qquad\qquad \left. + V_{x^*}^*(s_t, t) - \sum_{s_{t+1} \in \mathcal{S}} P(s_{t+1}|s_t, a_t, t) V_{x^*}^*(s_{t+1}, t+1) \right] \\
&= (1 - x^*) V_{\mathrm{sw}}^{\pi}(\hat{s}, h = 0) + V_{x^*}^*(\hat{s}, h = 0).
\end{aligned}
\tag{A.3}
$$

Subtracting the agent value under any policy $\pi$ from that under $\pi_{\mathrm{sw}}$ gives

$$
\begin{aligned}
V_A^{\pi, \Delta r^*}(\hat{s}, h = 0) &- V_A^{\pi_{\mathrm{sw}}, \Delta r^*}(\hat{s}, h = 0) \\
&= (1 - x^*)(V_{\mathrm{sw}}^{\pi}(\hat{s}, h = 0) - V_{\mathrm{sw}}^{\pi_{\mathrm{sw}}}(\hat{s}, h = 0)) \\
&\leq 0,
\end{aligned}
\tag{A.4}
$$

which implies $\pi_{\mathrm{sw}}$ can achieves the maximum agent value under $\Delta r^*$:

$$
V_A^{\pi_{\mathrm{sw}}, \Delta r^*}(\hat{s}, h = 0) = \overline{V}_A^{\Delta r^*}(\hat{s}, h = 0).
\tag{A.5}
$$

Then, we show (b):

$$
V_A^{\pi_{\mathrm{sw}}, \Delta r^*}(\hat{s}, h = 0) = V_A^{\pi_{\mathrm{sw}}, \Delta r_{\mathrm{sw}}}(\hat{s}, h = 0).
$$

Based on the definition of $\Delta r_{\mathrm{sw}}$, we can express $\Delta r_{\mathrm{sw}}$ for any triple $(s, a, h)$ as

$$
\Delta r_{\mathrm{sw}}(s, a, h) = \mathbf{1}_{\{x > 0\}}(\pi_{\mathrm{sw}}(a|s, h)) \Delta r(s, a, h),
$$

where $\mathbf{1}_{\{x>0\}}(x)$ is the indicator function:

$$\mathbf{1}_{\{x>0\}} = \begin{cases} 1, & x > 0, \\ 0, & x \leq 0. \end{cases}$$

By introducing the occupancy measure $\mu_{\mathrm{sw}}$ of policy $\pi_{\mathrm{sw}}$, which satisfies $\left(\sum_{a'} \mu_{\mathrm{sw}}(s, a', h)\right) \cdot \pi_{\mathrm{sw}}(a|s, h) = \mu_{\mathrm{sw}}(s, a, h)$ for any $(s, a, h)$, we have $\mathbf{1}_{\{x>0\}}(\pi_{\mathrm{sw}}(a|s, h)) \geq \mathbf{1}_{\{x>0\}}(\mu_{\mathrm{sw}}(s, a, h))$. Hence,

$$
\begin{aligned}
V_A^{\pi_{\mathrm{sw}}, \Delta r^*}(\hat{s}, h = 0) \\
= \sum_{s,a,h} \mu_{\mathrm{sw}}(s, a, h) \big( r_A(s, a, h) + \Delta r^*(s, a, h) \big) \\
= \sum_{s,a,h} \Big( \mu_{\mathrm{sw}}(s, a, h) r_A(s, a, h) + \mathbf{1}_{\{x>0\}}(\mu_{\mathrm{sw}}(s, a, h)) \cdot \mu_{\mathrm{sw}}(s, a, h) \Delta r^*(s, a, h) \Big) \\
\leq \sum_{s,a,h} \Big( \mu_{\mathrm{sw}}(s, a, h) r_A(s, a, h) + \mathbf{1}_{\{x>0\}}(\pi_{\mathrm{sw}}(a|s, h)) \cdot \mu_{\mathrm{sw}}(s, a, h) \Delta r^*(s, a, h) \Big) \\
= \sum_{s,a,h} \Big( \mu_{\mathrm{sw}}(s, a, h) r_A(s, a, h) + \mu_{\mathrm{sw}}(s, a, h) \Delta r_{\mathrm{sw}}(s, a, h) \Big) \\
= V_A^{\pi_{\mathrm{sw}}, \Delta r_{\mathrm{sw}}}(\hat{s}, h = 0).
\end{aligned}
$$

Meanwhile, since $\Delta r^*(s, a, h) \geq \Delta r_{\mathrm{sw}}(s, a, h)$ for any $(s, a, h)$, by construction, it follows that

$$V_A^{\pi_{\mathrm{sw}}, \Delta r^*}(\hat{s}, h = 0) \geq V_A^{\pi_{\mathrm{sw}}, \Delta r_{\mathrm{sw}}}(\hat{s}, h = 0).$$

Combining the above results, we can conclude that

$$V_A^{\pi_{\mathrm{sw}}, \Delta r^*}(\hat{s}, h = 0) = V_A^{\pi_{\mathrm{sw}}, \Delta r_{\mathrm{sw}}}(\hat{s}, h = 0).$$

Finally, we establish (c):

$$V_A^{\pi_{\mathrm{sw}}, \Delta r_{\mathrm{sw}}}(\hat{s}, h = 0) = \overline{V}_A^{\Delta r_{\mathrm{sw}}}(\hat{s}, h = 0).$$

To prove this, it suffices to show that for any policy $\pi$, $V_A^{\pi, \Delta r_{\mathrm{sw}}}(\hat{s}, h = 0) \leq V_A^{\pi_{\mathrm{sw}}, \Delta r_{\mathrm{sw}}}(\hat{s}, h = 0)$, which directly implies that $\pi_{\mathrm{sw}}$ attains the maximum agent value under $\Delta r_{\mathrm{sw}}$, i.e., $\overline{V}_A^{\Delta r_{\mathrm{sw}}}(\hat{s}, h = 0) = V_A^{\pi_{\mathrm{sw}}, \Delta r_{\mathrm{sw}}}(\hat{s}, h = 0)$.

The inequality, $V_A^{\pi, \Delta r_{\mathrm{sw}}}(\hat{s}, h = 0) \leq V_A^{\pi_{\mathrm{sw}}, \Delta r_{\mathrm{sw}}}(\hat{s}, h = 0)$, follows from the chain

$$V_A^{\pi, \Delta r_{\mathrm{sw}}}(\hat{s}, h = 0) \leq V_A^{\pi, \Delta r^*}(\hat{s}, h = 0) \leq V_A^{\pi_{\mathrm{sw}}, \Delta r^*}(\hat{s}, h = 0) = V_A^{\pi_{\mathrm{sw}}, \Delta r_{\mathrm{sw}}}(\hat{s}, h = 0).$$

The first inequality holds because the construction, $\Delta r_{\mathrm{sw}}(s, a, h) \leq \Delta r^*(s, a, h)$ holds for all $(s, a, h)$, which implies, for any action policy $\pi$, $V_A^{\pi, \Delta r_{\mathrm{sw}}}(\hat{s}, h = 0) \leq V_A^{\pi, \Delta r^*}(\hat{s}, h = 0)$. The second inequality follows from (A.4). The final equality holds because $\Delta r^*(s, a, h) = \Delta r_{\mathrm{sw}}(s, a, h)$ whenever $\pi_{\mathrm{sw}}(a|s, h) > 0$.

By combining (a), (b), and (c), we can conclude $\overline{V}_A^{\Delta r^*}(\hat{s}, h = 0) = \overline{V}_A^{\Delta r_{\mathrm{sw}}}(\hat{s}, h = 0)$ follows.

Next, we show that the principal's worst-case reward under $\Delta r_{\mathrm{sw}}$ is no worse than under $\Delta r^*$. Suppose an action policy $\pi$ is globally $\epsilon$-IC under $\Delta r_{\mathrm{sw}}$. By definition, this implies that the agent's value under $\Delta r_{\mathrm{sw}}$ achieves $\overline{V}_A^{\Delta r_{\mathrm{sw}}}(\hat{s}, h = 0) - \epsilon$. From the previous discussion, we know that $\overline{V}_A^{\Delta r^*}(\hat{s}, h = 0) = \overline{V}_A^{\Delta r_{\mathrm{sw}}}(\hat{s}, h = 0)$ and $\Delta r_{\mathrm{sw}} \leq \Delta r^*$. Consequently, the action policy $\pi$ is also globally $\epsilon$-IC under $\Delta r^*$. The converse, however, does not necessarily hold.

Thus, relative to $\Delta r^*$, the scheme $\Delta r_{\mathrm{sw}}$ reduces the set of agent policies that are globally $\epsilon$-IC. Moreover, for any given action policy, the principal's payoff under $\Delta r_{\mathrm{sw}}$ is at least as large as under $\Delta r^*$. It follows that the principal attains at least the same worst-case reward under $\Delta r_{\mathrm{sw}}$ as under $\Delta r^*$. Since $\Delta r^*$ is optimal, the principal's worst-case reward is identical under both schemes, thereby establishing the first part of the proposition.

**Action Policy under $\Delta r^*$**  To prove the latter part of Proposition 4.2, the key idea is to analyze the agent's response under a given subsidy scheme $\Delta r$. Recall from the preliminaries that we define the adversarial response $\pi_{\Delta r}$ to a subsidy scheme $\Delta r$ as

$$\pi_{\Delta r} \in \arg\min_{\pi \in \Pi(\Delta r)} V_P^{\pi, \Delta r}(\hat{s}, h = 0).$$

In the following discussion, we refer to $\pi_{\Delta r}$ as the *agent's adversarial action policy*. Further, we begin by analyzing the agent's behavior in a single-period instance and then extend the results to the multi-period case. Formally, when $H = 1$, there is only one state $\hat{s}$, so we can omit $(s, h)$ in the expressions. The optimization problem for agent then becomes

$$\min_{\pi} \sum_a \pi(a) r_P^{\Delta r}(a) \quad \text{s.t.} \sum_a \pi(a) r_A^{\Delta r}(a) \geq \max_a r_A^{\Delta r}(a) - \epsilon, \pi(a) \geq 0, \sum_a \pi(a) = 1.$$

As this is a linear program, we apply the KKT conditions to analyze the optimal solution. The Lagrangian function is

$$\begin{aligned}
\mathcal{L}(\pi; \alpha, \beta, V) &= \sum_a \pi(a) r_P^{\Delta r}(a) + \alpha \Big( \max_a r_A^{\Delta r}(a) - \epsilon - \sum_a \pi(a) r_A^{\Delta r}(a) \Big) \\
&\quad + \sum_a \beta(a)(-\pi(a)) + V \Big( 1 - \sum_a \pi(a) \Big) \\
&= \sum_a \pi(a) \Big( r_P^{\Delta r}(a) - \alpha r_A^{\Delta r}(a) - V + \beta(a) \Big) + V + \alpha \big( \max_a r_A^{\Delta r}(a) - \epsilon \big).
\end{aligned}$$

The resulting dual program is as followed:

$$\max_{\alpha, V} V + \alpha \big( \max_a r_A^{\Delta r}(a) - \epsilon \big) \quad \text{s.t.} \quad \alpha \geq 0, \ V \leq r_P(a) - \alpha r_A(a).$$

Let $\alpha^{\Delta r}$ and $V^{\Delta r}$ denote the optimal dual values under the subsidy scheme $\Delta r$, and let $\text{OTP}^{\Delta r}$ denote the final principal value under the same subsidy scheme. By complementary slackness, for any action $a$ such that

$$r_P^{\Delta r}(a) - \alpha^{\Delta r} r_A^{\Delta r}(a) = V^{\Delta r} = \min_a \big( r_P^{\Delta r}(a) - \alpha^{\Delta r} r_A^{\Delta r}(a) \big),$$

we have $\pi_{\Delta r}(a) \geq 0$. We refer to such actions as the *candidate actions* $a \in \overline{A}$, since they can potentially be chosen by the agent given subsidy scheme $\Delta r$.

However in certain problem instances, there exist candidate actions that do not appear in any agent's adversarial action policy. For example, consider $\epsilon = 1$ and two actions: the first action has principal reward $0$ and agent reward $1$, while the second action has principal reward $1$ and agent reward $2$. The unique agent's adversarial action policy deterministically selects the first action, yet setting $\alpha = 2$ would include both actions as candidate actions. In general, based on the value of optimal dual variable $\alpha^*$ under optimal subsidy scheme $\Delta r^*$, we claim there are three possible scenarios:

- **Case 1:** $\alpha^* = 0$. Every candidate action at this point attains the minimum principal value, so any action satisfying the globally $\epsilon$-IC constraint can be deterministically chosen.

- **Case 2:** $\alpha^* > 0$, and all candidate actions satisfy the globally $\epsilon$-IC constraint. Complementary slackness implies the agent's value is exactly $\max_a \overline{r}_A(a) - \epsilon$, so only actions attaining this value can be chosen. This case coincides with the above example.

- **Case 3:** $\alpha^* > 0$, and some candidate actions have agent reward below $\max_a \overline{r}_A(a) - \epsilon$. Then, the agent can mix actions above and below this threshold to form an agent's adversarial action policy, leading to the fact that any candidate action may become part of agent response.

To further explain the agent's behavior pattern in **Case 2** and **Case 3**, we use the following equations to show that when $\alpha^{\Delta r} > 0$, as long as the action policy distribution $\pi$ is supported only on candidate actions and achieves an agent value exactly equal to $\max_a \overline{r}_A(a) - \epsilon$, the policy $\pi$ constitutes one possible adversarial action policy of the agent. In other words, we only need to consider how

to organize the policy distribution supported on candidate actions so as to achieve an agent value exactly equal to $\max_a \bar{r}_A(a) - \epsilon$.

$$\sum_a \pi(a) r_P^{\Delta r}(a) = \sum_a \pi(a)(r_P^{\Delta r}(a) - \alpha^{\Delta r} r_A^{\Delta r}(a)) + \alpha^{\Delta r} \sum_a \pi(a) r_A^{\Delta r}(a)$$

$$= \sum_a \pi(a) V^{\Delta r} + \alpha \Delta r (\max_a r_A^{\Delta r}(a) - \epsilon)$$

$$= \mathrm{OPT}^{\Delta r}$$

We now extend to the multi-period case. Observe that under subsidy scheme $\Delta r$, any multi-period policy $\pi_i$ can be viewed as a single-period action $a_i$ with principal reward $V_P^{\pi_i, \Delta r}(\hat{s}, h = 0)$ and agent reward $V_A^{\pi_i, \Delta r}(\hat{s}, h = 0)$, yielding a single-period instance with infinitely many actions. Although this transformation is generally intractable, it provides a useful framework for analyzing the properties of action policies. Suppose the single-period agent' adversarial action policy is $\pi_s^*$, the multi-period agent's adversarial action policy can be recovered as $\pi^*(a|s, h) = \sum_i \pi_s^*(a_i) \pi_i(a|s, h)$ for any state $s$, action $a$ and timestep $h$, where $\pi_s^*(a_i)$ denotes the probability assigned to policy $\pi_i$. Accordingly, we define the *candidate policy* $\overline{\pi} \in \overline{\Pi}$ under subsidy scheme $\Delta r$ as

$$V_P^{\overline{\pi}, \Delta r}(\hat{s}, h = 0) - \alpha^{\Delta r} V_A^{\overline{\pi}, \Delta r}(\hat{s}, h = 0) = \min_\pi \left\{ V_P^{\pi, \Delta r}(\hat{s}, h = 0) - \alpha^{\Delta r} V_A^{\pi, \Delta r}(\hat{s}, h = 0) \right\},$$

analogous to the single-period candidate actions, so that the adversarial agent's action policy can be expressed as a convex combination of these candidate policies.

We next establish that, under the optimal subsidy scheme $\Delta r^*$ from Theorem 4.1, **every** action policy $\pi$, including $\pi_{\mathrm{sw}}$, qualifies as a candidate. Recall from (4.3) that

$$\Delta r^*(s, a, h) = V_{x^*}^*(s, h) - Q_{x^*}^*(s, a, h)$$

$$= V_{x^*}^*(s, h) - x^* r_{\mathrm{sw}}(s, a, h) + r_P(s, a, h) - \sum_{s' \in \mathcal{S}} P(s'|s, a, h) V_{x^*}^*(s', h + 1)$$

$$= (1 - x^*) r_P(s, a, h) - x^* r_A(s, a, h) + V_{x^*}^*(s, h) - \sum_{s' \in \mathcal{S}} P(s'|s, a, h) V_{x^*}^*(s', h + 1),$$

where $x^* = \frac{\alpha^*}{1 + \alpha^*}$. For any action policy, under optimal subsidy scheme $\Delta r^*$ substituting $\Delta r^*$ into the expression $V_P^{\pi, \Delta r}(\hat{s}, h = 0) - \alpha^* V_A^{\pi, \Delta r}(\hat{s}, h = 0)$ yields

$$V_P^{\pi, \Delta r^*}(\hat{s}, h = 0) - \alpha^* V_A^{\pi, \Delta r^*}(\hat{s}, h = 0)$$

$$= \mathbb{E}_\pi \left[ \sum_{t=0}^{H-1} r_P(s_t, a_t, t) - \alpha^* r_A(s_t, a_t, t) - (1 + \alpha^*) \Delta r^*(s_t, a_t, t) \right]$$

$$= \mathbb{E}_\pi \left[ \sum_{t=0}^{H-1} r_P(s_t, a_t, t) - \frac{x^*}{1 - x^*} r_A(s_t, a_t, t) - \frac{1}{1 - x^*} \Delta r^*(s_t, a_t, t) \right]$$

$$= \mathbb{E}_\pi \left[ \sum_{t=0}^{H-1} r_P(s_t, a_t, t) - \frac{x^*}{1 - x^*} r_A(s_t, a_t, t) \right. \tag{A.6}$$

$$- \frac{1}{1 - x^*} \left( (1 - x^*) r_P(s_t, a_t, t) - x^* r_A(s_t, a_t, t) \right)$$

$$\left. - \frac{1}{1 - x^*} \left( V_{x^*}^*(s_t, t) - \sum_{s_{t+1} \in \mathcal{S}} P(s_{t+1}|s_t, a_t, t + 1) V_{x^*}^*(s_{t+1}, t + 1) \right) \right]$$

$$= - \frac{1}{1 - x^*} V_{x^*}^*(\hat{s}, h = 0).$$

Thus, every action policy $\pi$ yields the same value

$$V_P^{\pi, \Delta r^*}(\hat{s}, h = 0) - \alpha^* V_A^{\pi, \Delta r^*}(\hat{s}, h = 0).$$

Consequently, all action policies qualify as candidate policies, and we have

$$\min_\pi \left\{ V_P^{\pi, \Delta r^*}(\hat{s}, h = 0) - \alpha^* V_A^{\pi, \Delta r^*}(\hat{s}, h = 0) \right\} = -\frac{1}{1 - x^*} V_{x^*}^*(\hat{s}, h = 0).$$

Having established this, we note that not all candidate policies necessarily receive positive probability in the support of an agent's adversarial action policy, as illustrated in the single-period analysis. Similarly, we distinguish cases based on the value of the optimal dual variable $\alpha^*$: when $\alpha^* = 0$, the corresponding optimal solution is $x^* = 0$, making the proposition trivial; When $\alpha^* > 0$, we assert that **Case 2** will never happen under optimal subsidy scheme $\Delta r^*$ as there exists a candidate policy $\hat{\pi}$ whose agent value is strictly smaller than $\overline{V}_A^{\Delta r^*}(\hat{s}, h = 0) - \epsilon$. Moreover, this policy $\hat{\pi}$ can be combined with $\pi_{sw}$ to construct the agent's adversarial action policy.

In detail, we construct the candidate policy $\hat{\pi}$ by analyzing the derivative of the objective function. Applying the envelope theorem (Milgrom & Segal, 2002), the derivative of the objective with respect to $x$ in Theorem 4.1 is given by

$$F'(x) = V_{sw}^*(\hat{s}, h = 0) - V_{sw}^{\pi_x}(\hat{s}, h = 0) - \frac{\epsilon}{(1 - x)^2}, \tag{A.7}$$

where $\pi_x = \arg\max_\pi \{ x V_{sw}^\pi(\hat{s}, h = 0) - V_P^{\pi, \Delta r = 0}(\hat{s}, h = 0) \}$. In general, the objective function may have a finite number of non-differentiable points, arising from the potential non-uniqueness of $V_{sw}^{\pi_x}(\hat{s}, h = 0)$. Nevertheless, since the set of sub-differentials can be fully characterized by the derivative expression, for simplicity and clarity we do not distinguish between derivatives and sub-derivatives, and we treat stationary points by directly setting $F'(x) = 0$.

Since the objective function is concave, the optimal solution $x^*$ and the corresponding policy $\pi_{x^*}$ can be characterized by the vanishing derivative condition. Furthermore, the requirement $\alpha^* > 0$ ensures that $x^* \in (0, 1)$, implying that $x^*$, as an interior optimum, necessarily exists as a stationary point. Consequently, imposing $F'(x) = 0$ yields

$$V_{sw}^*(\hat{s}, h = 0) - V_{sw}^{\pi_{x^*}}(\hat{s}, h = 0) = \frac{\epsilon}{(1 - x^*)^2}.$$

In general, $\pi_{x^*}$ can be represented as any convex combination of some action policies $\widetilde{\pi}_x$ maximizing $x V_{sw}^{\widetilde{\pi}_x}(\hat{s}, h = 0) - V_P^{\widetilde{\pi}_x, \Delta r = 0}(\hat{s}, h = 0)$. Since every action policy qualifies as a candidate policy under $\Delta r^*$, $\pi_{x^*}$ can equivalently be viewed as a convex combination of candidate policies. Hence, there exists a candidate policy $\hat{\pi}$ such that

$$\hat{\pi} = \arg\max_\pi \{ x V_{sw}^\pi(\hat{s}, h = 0) - V_P^{\pi, \Delta r = 0}(\hat{s}, h = 0) \},$$

with

$$V_{sw}^*(\hat{s}, h = 0) - V_{sw}^{\hat{\pi}}(\hat{s}, h = 0) \geq \frac{\epsilon}{(1 - x^*)^2}.$$

To upper bound the agent value of $\hat{\pi}$, using equations (A.3) and (A.5), we deduce that for action policy $\hat{\pi}$,

$$\overline{V}_A^{\Delta r^*}(\hat{s}, h = 0) - V_A^{\hat{\pi}, \Delta r^*}(\hat{s}, h = 0) = (1 - x^*) \left( V_{sw}^*(\hat{s}, h = 0) - V_{sw}^{\hat{\pi}}(\hat{s}, h = 0) \right), \tag{A.8}$$

which, combined with the preceding inequality, implies

$$\overline{V}_A^{\Delta r^*}(\hat{s}, h = 0) - V_A^{\hat{\pi}, \Delta r^*}(\hat{s}, h = 0) \geq \frac{\epsilon}{1 - x^*} \geq \epsilon.$$

Thus, by mixing $\pi_{sw}$ and $\hat{\pi}$ with weight $p$, we construct an agent's adversarial action policy whose value is exactly $\overline{V}_A^{\Delta r^*}(\hat{s}, h = 0) - \epsilon$. Moreover, the mixing weight on $\pi_{sw}$ must satisfy $p \geq x^*$. The derivation of this lower bound on $p$ is as follows:

First, by the definition of the mixed policy and dual variable $\alpha^* > 0$,

$$p V_A^{\pi_{sw}, \Delta r^*}(\hat{s}, h = 0) + (1 - p) V_A^{\hat{\pi}, \Delta r^*}(\hat{s}, h = 0) = \overline{V}_A^{\Delta r^*}(\hat{s}, h = 0) - \epsilon.$$

Since $V_A^{\pi_{\mathrm{sw}}, \Delta r^*}(\hat{s}, h = 0) = \overline{V}_A^{\Delta r^*}(\hat{s}, h = 0)$, this equality can be rewritten as

$$p\epsilon + (1 - p)\Big(V_A^{\hat{\pi}, \Delta r^*}(\hat{s}, h = 0) - \big(\overline{V}_A^{\Delta r^*}(\hat{s}, h = 0) - \epsilon\big)\Big) = 0.$$

Next, using the inequality $\overline{V}_A^{\Delta r^*}(\hat{s}, h = 0) - V_A^{\hat{\pi}, \Delta r^*}(\hat{s}, h = 0) \geq (1 - x^*)^{-1}\epsilon$, we obtain

$$p\epsilon + (1 - p)\big(\frac{-\epsilon}{1 - x^*} + \epsilon\big) \geq 0.$$

Finally, dividing both sides by $\epsilon > 0$ gives $p \geq x^*$.

**Action Policy under $\Delta r_{\mathrm{sw}}$**   When the optimal subsidy scheme is shifted from $\Delta r^*$ to $\Delta r_{\mathrm{sw}}$, our primary objective—towards establishing the latter part of the proposition—is to verify that both $\hat{\pi}$ and $\pi_{\mathrm{sw}}$ continue to satisfy the requirements of candidate policies. We first claim that the optimal solution $x^*$ and the optimal dual variable $\alpha^*$ remain unchanged under this modification. By the first part of Proposition 4.2, the principal's optimal value is preserved in this transition. Furthermore, recalling from Theorem 4.1 that the objective function $F(x)$ is concave, it follows that both $x^*$ and $\alpha^*$ remain optimal.

Then, to determine whether an action policy $\overline{\pi}$, such as $\hat{\pi}$ and $\pi_{\mathrm{sw}}$, qualifies as a candidate policy under $\Delta r_{\mathrm{sw}}$, it suffices to verify whether

$$\begin{aligned}
&V_P^{\overline{\pi}, \Delta r_{\mathrm{sw}}}(\hat{s}, h = 0) - \alpha^* V_A^{\overline{\pi}, \Delta r_{\mathrm{sw}}}(\hat{s}, h = 0) \\
&= \min_\pi \Big\{V_P^{\overline{\pi}, \Delta r_{\mathrm{sw}}}(\hat{s}, h = 0) - \alpha^* V_A^{\pi, \Delta r_{\mathrm{sw}}}(\hat{s}, h = 0)\Big\}.
\end{aligned} \tag{A.9}$$

For clarity, we define

$$g(\pi; \Delta r) \triangleq V_P^{\pi, \Delta r}(\hat{s}, h = 0) - \alpha^* V_A^{\pi, \Delta r}(\hat{s}, h = 0).$$

The proof proceeds in two steps. First, we establish a lower bound for the right-hand side of (A.9) as

$$\min_\pi g(\pi; \Delta r_{\mathrm{sw}}) = \min_\pi \Big\{V_P^{\pi, \Delta r_{\mathrm{sw}}}(\hat{s}, h = 0) - \alpha^* V_A^{\pi, \Delta r_{\mathrm{sw}}}(\hat{s}, h = 0)\Big\} \geq -\frac{1}{1 - x^*} V_{x^*}^*(\hat{s}, h = 0).$$

Second, we show that for the action policies $\hat{\pi}$ and $\pi_{\mathrm{sw}}$,

$$g(\hat{\pi}; \Delta r_{\mathrm{sw}}) = g(\pi_{\mathrm{sw}}; \Delta r_{\mathrm{sw}}) = -\frac{1}{1 - x^*} V_{x^*}^*(\hat{s}, h = 0).$$

In the first step, note that since $\Delta r^* \geq \Delta r_{\mathrm{sw}}$, we have for any action policy $\pi$,

$$V_P^{\pi, \Delta r^*}(\hat{s}, h = 0) \leq V_P^{\pi, \Delta r_{\mathrm{sw}}}(\hat{s}, h = 0), \quad V_A^{\pi, \Delta r^*}(\hat{s}, h = 0) \geq V_A^{\pi, \Delta r_{\mathrm{sw}}}(\hat{s}, h = 0).$$

As $\alpha^* \geq 0$, it follows that for any $\pi$,

$$\begin{aligned}
g(\pi, \Delta r_{\mathrm{sw}}) &= \min_\pi \Big\{V_P^{\pi, \Delta r_{\mathrm{sw}}}(\hat{s}, h = 0) - \alpha^* V_A^{\pi, \Delta r_{\mathrm{sw}}}(\hat{s}, h = 0)\Big\} \\
&\geq \min_\pi \Big\{V_P^{\pi, \Delta r^*}(\hat{s}, h = 0) - \alpha^* V_A^{\pi, \Delta r^*}(\hat{s}, h = 0)\Big\} \\
&= -\frac{1}{1 - x^*} V_{x^*}^*(\hat{s}, h = 0).
\end{aligned} \tag{A.10}$$

In the second step, to prove $\hat{\pi}$ is a candidate policy and attains the minimum $-\frac{1}{1 - x^*} V_{x^*}^*(\hat{s}, h = 0)$, we connect $g(\hat{\pi}; \Delta r_{\mathrm{sw}})$ and $-\frac{1}{1 - x^*} V_{x^*}^*(\hat{s}, h = 0)$ via $g(\hat{\pi}; \Delta r = 0)$. Substituting $x^* = \frac{\alpha^*}{1 + \alpha^*}$, we

obtain

$$
\begin{aligned}
g(\hat{\pi}; \Delta r = 0) &= V_P^{\hat{\pi}, \Delta r = 0}(\hat{s}, h = 0) - \alpha^* V_A^{\hat{\pi}, \Delta r = 0}(\hat{s}, h = 0) \\
&= V_P^{\hat{\pi}, \Delta r = 0}(\hat{s}, h = 0) - \frac{x^*}{1 - x^*} V_A^{\hat{\pi}, \Delta r = 0}(\hat{s}, h = 0) \\
&= -\frac{1}{1 - x^*} \left( (x^* - 1) V_P^{\hat{\pi}, \Delta r = 0}(\hat{s}, h = 0) + x^* V_A^{\hat{\pi}, \Delta r = 0}(\hat{s}, h = 0) \right) \\
&= -\frac{1}{1 - x^*} \left( x^* V_{\mathrm{sw}}^{\hat{\pi}}(\hat{s}, h = 0) - V_P^{\hat{\pi}, \Delta r = 0}(\hat{s}, h = 0) \right) \\
&= -\frac{1}{1 - x^*} \max_\pi \left\{ x^* V_{\mathrm{sw}}^{\pi}(\hat{s}, h = 0) - V_P^{\pi, \Delta r = 0}(\hat{s}, h = 0) \right\} \qquad (A.11) \\
&= \min_\pi \left\{ V_P^{\pi, \Delta r = 0}(\hat{s}, h = 0) - \frac{x^*}{1 - x^*} V_A^{\pi, \Delta r = 0}(\hat{s}, h = 0) \right\} \\
&= -\frac{1}{1 - x^*} V_{x^*}^*(\hat{s}, h = 0). \qquad (A.12)
\end{aligned}
$$

Equality (A.11) follows since

$$
\hat{\pi} = \arg\max_\pi \{ x^* V_{\mathrm{sw}}^{\pi}(\hat{s}, h = 0) - V_P^{\pi, \Delta r = 0}(\hat{s}, h = 0) \},
$$

while (A.12) corresponds to the definition of $V_x^*(\hat{s}, h = 0)$ in Theorem 4.1:

$$
V_x^*(\hat{s}, h = 0) \triangleq -(1 - x) \cdot \min_\pi \left\{ V_P^{\pi, \Delta r = 0}(\hat{s}, h = 0) - \frac{x}{1 - x} V_A^{\pi, \Delta r = 0}(\hat{s}, h = 0) \right\}.
$$

We next turn to discuss the equality between $g(\hat{\pi}, \Delta r_{\mathrm{sw}})$ and $g(\hat{\pi}, \Delta r = 0)$. Combining (A.10) with $g(\hat{\pi}, \Delta r = 0) = -\frac{1}{1 - x^*} V_{x^*}^*(\hat{s}, h = 0)$, we obtain

$$
g(\hat{\pi}, \Delta r_{\mathrm{sw}}) \geq g(\hat{\pi}, \Delta r = 0).
$$

To establish equality, it suffices to show $g(\hat{\pi}, \Delta r_{\mathrm{sw}}) \leq g(\hat{\pi}, \Delta r = 0)$. Since $\Delta r_{\mathrm{sw}}(s, a, h) \geq \Delta r(s, a, h)$ for all $(s, a, h)$, we have

$$
V_P^{\hat{\pi}, \Delta r_{\mathrm{sw}}}(\hat{s}, h = 0) \leq V_P^{\hat{\pi}, \Delta r = 0}(\hat{s}, h = 0), \quad V_A^{\hat{\pi}, \Delta r_{\mathrm{sw}}}(\hat{s}, h = 0) \geq V_A^{\hat{\pi}, \Delta r = 0}(\hat{s}, h = 0).
$$

With $\alpha^* \geq 0$, it follows that

$$
\begin{aligned}
g(\hat{\pi}, \Delta r_{\mathrm{sw}}) &= V_P^{\hat{\pi}, \Delta r_{\mathrm{sw}}}(\hat{s}, h = 0) - \alpha^* V_A^{\hat{\pi}, \Delta r_{\mathrm{sw}}}(\hat{s}, h = 0) \\
&\leq V_P^{\hat{\pi}, \Delta r = 0}(\hat{s}, h = 0) - \alpha^* V_A^{\hat{\pi}, \Delta r = 0}(\hat{s}, h = 0) \\
&= g(\hat{\pi}, \Delta r = 0).
\end{aligned}
$$

Thus $g(\hat{\pi}, \Delta r_{\mathrm{sw}}) = g(\hat{\pi}, \Delta r = 0)$.

For $\pi_{\mathrm{sw}}$, recall that after changing from $\Delta r^*$ to $\Delta r_{\mathrm{sw}}$, its agent value remains unchanged. Since the social welfare is unaffected by the subsidy scheme, the principal's value also remains the same. Hence,

$$
\begin{aligned}
g(\pi_{\mathrm{sw}}, \Delta r_{\mathrm{sw}}) &= V_P^{\pi_{\mathrm{sw}}, \Delta r_{\mathrm{sw}}}(\hat{s}, h = 0) - \alpha^* V_A^{\pi_{\mathrm{sw}}, \Delta r_{\mathrm{sw}}}(\hat{s}, h = 0) & (A.13) \\
&= V_P^{\pi_{\mathrm{sw}}, \Delta r^*}(\hat{s}, h = 0) - \alpha^* V_A^{\pi_{\mathrm{sw}}, \Delta r^*}(\hat{s}, h = 0) & (A.14) \\
&= -\frac{1}{1 - x^*} V_{x^*}^*(\hat{s}, h = 0), & (A.15)
\end{aligned}
$$

where the last equality follows from (A.6).

Finally, to bound the probability weight, recall that both the maximum agent value and the agent value of $\pi_{\mathrm{sw}}$ remain unchanged, i.e.

$$
V_A^{\pi_{\mathrm{sw}}, \Delta r_{\mathrm{sw}}}(\hat{s}, h = 0) = V_A^{\pi_{\mathrm{sw}}, \Delta r^*}(\hat{s}, h = 0) = \overline{V}_A^{\Delta r_{\mathrm{sw}}}(\hat{s}, h = 0) = \overline{V}_A^{\Delta r^*}(\hat{s}, h = 0).
$$

Meanwhile, for $\hat{\pi}$, its deviation from the maximum agent value is still bounded as

$$
\begin{aligned}
V_A^{\hat{\pi}, \Delta r_{\text{sw}}}(\hat{s}, h = 0) &\leq V_A^{\hat{\pi}, \Delta r^*}(\hat{s}, h = 0) \\
&\leq \overline{V}_A^{\Delta r^*}(\hat{s}, h = 0) - \frac{\epsilon}{1 - x^*} \\
&= \overline{V}_A^{\Delta r_{\text{sw}}}(\hat{s}, h = 0) - \frac{\epsilon}{1 - x^*}.
\end{aligned}
$$

Therefore, by the same reasoning as under $\Delta r^*$, we conclude that $\pi_{\text{sw}}$ and $\hat{\pi}$ constitute an optimal adversarial pair for the agent, with $\pi_{\text{sw}}$ assigned a probability weight of at least $x^*$.

## A.6 PROOF OF PROPOSITION 4.3

According to the proof Theorem 4.1, recall that the optimal subsidy scheme $\Delta r^*$ given optimal dual variable $(\alpha^*, V^*)$ is

$$
\Delta r^*(s, a, h) = \frac{1}{1 + \alpha^*} \left( -V^*(s, h) + \sum_{s' \in \mathcal{S}} P(s'|s, a, h) V^*(s', h+1) + r_P(s, a, h) - \alpha r_A(s, a, h) \right).
$$

Therefore, the agent's action policy $\pi_{\Delta r^*}$ under $\Delta r^*$ satisfies

$$
V_P^{\pi_{\Delta r^*}, \Delta r^*}(\hat{s}, h = 0) = \frac{1}{1 + \alpha^*} V^*(\hat{s}, h = 0) + \frac{\alpha^*}{1 + \alpha^*} V_{\text{sw}}^*(\hat{s}, h = 0) - \alpha^* \epsilon,
$$

$$
V_A^{\pi_{\Delta r^*}, \Delta r^*}(\hat{s}, h = 0) \geq \overline{V}_A^{\Delta r^*}(\hat{s}, h = 0) - \epsilon = \frac{1}{1 + \alpha^*} (V_{\text{sw}}^*(\hat{s}, h = 0) - V^*(\hat{s}, h = 0)) - \epsilon.
$$

Based on the KKT condition, when $x^* \in (0, 1)$ hence $\alpha^* > 0$, the above inequality becomes the equality. Therefore, combining the above two equations, we can conclude that

$$
\delta_{\text{sw}} = V_{\text{sw}}^*(\hat{s}, h = 0) - V_{\text{sw}}^{\pi_{\Delta r^*}}(\hat{s}, h = 0) = (1 + \alpha^*)\epsilon = \frac{\epsilon}{1 - x^*}.
$$

The key to analyze such gap is to examine the relationship between $\epsilon$ and $x^*$. By applying the envelope theorem (Milgrom & Segal, 2002), the derivative of the objective function respect to $x$ in Theorem 4.1 is

$$
F'(x) = V_{\text{sw}}^*(\hat{s}, h = 0) - V_{\text{sw}}^{\pi_x}(\hat{s}, h = 0) - \frac{\epsilon}{(1 - x)^2}, \tag{A.16}
$$

where $\pi_x = \arg\max_\pi \{ x V_{\text{sw}}^\pi(\hat{s}, h = 0) - V_P^{\pi, \Delta r = 0}(\hat{s}, h = 0) \}$. In general, the objective function contains a finite number of non-differentiable points, which arise from the non-uniqueness of $V_{\text{sw}}^{\pi_x}(\hat{s}, h = 0)$. Nevertheless, since the set of sub-differential can be fully characterized by the expression of the derivative, for the sake of simplicity and clarity we do not distinguish between derivatives and sub-derivatives, and in the discussion of stationary points we directly set the derivative $F'(x)$ to 0.

Meanwhile, since the objective function is concave, the optimal solution $x^*$ and the corresponding policy $\pi_{x^*}$ can be characterized by the condition that the derivative vanishes. Moreover, the requirement $\alpha^* > 0$ ensures that $x^* \in (0, 1)$, which implies that $x^*$, as an interior optimum, necessarily exists as a stationary point. Consequently, imposing the condition $F'(x) = 0$ yields

$$
1 - x^* = \sqrt{\frac{\epsilon}{V_{\text{sw}}^*(\hat{s}, h = 0) - V_{\text{sw}}^{\pi_{x^*}}(\hat{s}, h = 0)}}. \tag{A.17}
$$

Since $\pi_{x^*}$ is an action policy, we can immediately have a trivial bound as

$$
V_{\text{sw}}^*(\hat{s}, h = 0) - V_{\text{sw}}^{\pi_{x^*}}(\hat{s}, h = 0) \geq V_{\text{sw}}^*(\hat{s}, h = 0) - \min_\pi V_{\text{sw}}^\pi(\hat{s}, h = 0),
$$

However, in fact, we can obtain a tighter constant bound on $V_{\text{sw}}^*(\hat{s}, h = 0) - V_{\text{sw}}^{\pi_{x^*}}(\hat{s}, h = 0)$. Denote $\pi_A$ as the action policy that attains the minimum principal value in the absence of a subsidy, i.e.,

$$
\pi_A = \arg\min_\pi V_P^{\pi, \Delta r = 0}(\hat{s}, h = 0),
$$

and if multiple action policies achieve this minimum, we select $\pi_A$ to be the one among them that maximizes social welfare. We claim that

$$
V_{\text{sw}}^*(\hat{s}, h = 0) - V_{\text{sw}}^{\pi_{x^*}}(\hat{s}, h = 0) \leq V_{\text{sw}}^*(\hat{s}, h = 0) - V_{\text{sw}}^{\pi_A}(\hat{s}, h = 0).
$$

To prove this, we show that for any $x \in [0, 1)$,

$$V_{\text{sw}}^{\pi_x}(\hat{s}, h = 0) \geq V_{\text{sw}}^{\pi_A}(\hat{s}, h = 0).$$

Recall that

$$\pi_x = \arg\max_{\pi} \{ x V_{\text{sw}}^{\pi}(\hat{s}, h = 0) - V_P^{\pi, \Delta r = 0}(\hat{s}, h = 0) \}.$$

By simple rearrangement and using $x > 0$, we obtain

$$V_{\text{sw}}^{\pi_x}(\hat{s}, h = 0) - V_{\text{sw}}^{\pi_A}(\hat{s}, h = 0) \geq \frac{1}{x} \left( V_P^{\pi_x, \Delta r = 0}(\hat{s}, h = 0) - V_P^{\pi_A, \Delta r = 0}(\hat{s}, h = 0) \right).$$

As $\pi_A$ minimizes the principal value with $\Delta r = 0$, the right-hand side is nonnegative, which immediately implies

$$V_{\text{sw}}^{\pi_x}(\hat{s}, h = 0) \geq V_{\text{sw}}^{\pi_A}(\hat{s}, h = 0),$$

as desired. Moreover, this bound is tight: based on the definition of $\pi_A$, there always exists some $x_0 \in (0, 1)$ such that for any $x \in [0, x_0]$, the policy $\pi_A$ maximizes $x V_{\text{sw}}^{\pi}(s, h) - V_P^{\pi, \Delta r = 0}(s, h)$.

Having established a bound on $V_{\text{sw}}^{*}(\hat{s}, h = 0) - V_{\text{sw}}^{\pi_{x^*}}(\hat{s}, h = 0)$, we can immediately derive a corresponding bound on $1 - x^*$:

$$1 - x^* \leq \sqrt{\frac{\epsilon}{V_{\text{sw}}^{*}(\hat{s}, h = 0) - V_{\text{sw}}^{\pi_A}(\hat{s}, h = 0)}}.$$

Finally, applying the condition $F'(x) = 0$ once more, we obtain

$$\delta_{\text{sw}} = \frac{\epsilon}{1 - x^*} = \left( V_{\text{sw}}^{*}(\hat{s}, h = 0) - V_{\text{sw}}^{\pi_A}(\hat{s}, h = 0) \right) (1 - x^*)$$

$$\leq \sqrt{\left( V_{\text{sw}}^{*}(\hat{s}, h = 0) - V_{\text{sw}}^{\pi_A}(\hat{s}, h = 0) \right) \cdot \epsilon},$$

demonstrating that the social welfare gap $\delta_{\text{sw}}$ is upper bounded by $O(\sqrt{\epsilon})$.

### A.6.1 THE DEPENDENCE OF $\delta_{\text{sw}}$ ON $\epsilon$

When analyzing the relationship between $\delta_{\text{sw}}$ and $\epsilon$, a straightforward observation is that the effectiveness of the limited subsidy diminishes as $\epsilon$ approaches infinity, allowing the agent to bypass the globally $\epsilon$-IC constraint and achieve a trivial minimization of the leader's value. In this scenario, even as $\epsilon$ continues to increase, the social welfare gap remains unchanged. Therefore, our analysis focuses on relatively small values of $\epsilon$, examining how the social welfare gap grows $\delta_{\text{sw}}$ as $\epsilon$ increases and $x^* \in (0, 1)$.

The key issue in equation (A.17) is that changes in $\epsilon$ may simultaneously affect both $x^*$ and the term $V_{\text{sw}}^{*}(\hat{s}, h = 0) - V_{\text{sw}}^{\pi_{x^*}}(\hat{s}, h = 0)$. However, a closer analysis of the derivative reveals that $\epsilon$ cannot influence these two quantities simultaneously. Before presenting the detailed argument, we first establish the following lemma, which shows that $x^*$ is monotone in $\epsilon$.

**Lemma A.1.** *The optimal solution $x^* \in (0, 1)$ is monotonically non-increasing in $\epsilon$.*

*Proof.* Recall that

$$V_x^{*}(\hat{s}, h = 0) \triangleq \max_{\pi} \left\{ x V_{\text{sw}}^{\pi}(\hat{s}, h = 0) - V_P^{\pi, \Delta r = 0}(\hat{s}, h = 0) \right\}.$$

Since $V_x^{*}(\hat{s}, h = 0)$ is the maximum of finitely many linear functions in $x$, its derivative $V_{\text{sw}}^{\pi_x}(\hat{s}, h = 0)$ is non-decreasing in $x$. Consequently, $V_{\text{sw}}^{*}(\hat{s}, h = 0) - V_{\text{sw}}^{\pi_{x^*}}(\hat{s}, h = 0)$ is non-increasing in $x$. Further, for $x^* \in (0, 1)$, imposing the stationarity condition $F'(x) = 0$ yields

$$\epsilon = (1 - x^*)^2 \left( V_{\text{sw}}^{*}(\hat{s}, h = 0) - V_{\text{sw}}^{\pi_{x^*}}(\hat{s}, h = 0) \right).$$

As $x^*$ increases, both $(1 - x^*)$ and $V_{\text{sw}}^{*}(\hat{s}, h = 0) - V_{\text{sw}}^{\pi_{x^*}}(\hat{s}, h = 0)$ decrease, making the right-hand side of the above equation monotonically non-increasing in $x^*$. Thus, for the equality to hold, an increase in $\epsilon$ must be matched by an non-increase in $x^*$, which proves the claim. $\square$

We then analyze the two possible cases of $x^*$ given $\epsilon$, corresponding to $\delta_{\text{sw}}$ scaling as $O(\sqrt{\epsilon})$ or $O(\epsilon)$, respectively:

- When the piecewise linear function $V_x^*(s, h)$ is differentiable at $x^*$, every $\pi_{x^*} \in \Pi_{x^*}$ yields the same social welfare $V_{\text{sw}}^{\pi_{x^*}}(\hat{s}, h = 0)$. Owing to this uniqueness, there exists a small interval $\delta$ such that for all $x \in (x^* - \delta, x^* + \delta)$, the policy $\pi_{x^*}$ remains unchanged. Consequently, $x^*$ is linearly related to $\sqrt{\epsilon}$, and the social welfare gap scales as $O(\sqrt{\epsilon})$.

- When $V_x^*(s, h)$ is non-differentiable at $x^*$, different $\pi_{x^*}$ induce different levels of social welfare. In this case as $F(x)$ is concave, within $\Pi_{x^*}$, there exist two policies $\pi_{x^*}^l$ and $\pi_{x^*}^r$ achieving the minimum and maximum social welfare, respectively, which define the left- and right-hand derivatives around $x^*$. We claim as long as

$$V_{\text{sw}}^*(\hat{s}, h = 0) - V_{\text{sw}}^{\pi_{x^*}^l}(\hat{s}, h = 0) \geq \frac{\epsilon'}{(1 - x^*)^2} \geq V_{\text{sw}}^*(\hat{s}, h = 0) - V_{\text{sw}}^{\pi_{x^*}^r}(\hat{s}, h = 0),$$

the optimal solution for $\epsilon'$ remains $x^*$, and the social welfare gap scales as $O(\epsilon)$. To see this, consider the counterexample. Recall from the proof of Lemma A.1 that $V_{\text{sw}}^{\pi_{x^*}}(\hat{s}, h = 0)$ is non-decreasing in $x$. Suppose the solution for $\epsilon'$ is $x'$. If $x' < x^*$, then

$$\begin{aligned} F'(x') &= V_{\text{sw}}^*(\hat{s}, h = 0) - V_{\text{sw}}^{\pi_{x'}}(\hat{s}, h = 0) - \frac{\epsilon}{(1 - x')^2} \\ &\geq V_{\text{sw}}^*(\hat{s}, h = 0) - V_{\text{sw}}^{\pi_{x^*}^l}(\hat{s}, h = 0) - \frac{\epsilon}{(1 - x')^2} \\ &> V_{\text{sw}}^*(\hat{s}, h = 0) - V_{\text{sw}}^{\pi_{x^*}^l}(\hat{s}, h = 0) - \frac{\epsilon}{(1 - x^*)^2} \\ &\geq 0, \end{aligned}$$

which shows $x'$ is not a stationary point, and hence not optimal for a concave function. The proof for $x' > x^*$ follows analogously by replacing $\pi_{x^*}^l$ with $\pi_{x^*}^r$ and showing $F'(x') < 0$.

## A.7 Omitted Calculations and Proof in Section 5

### A.7.1 Technical Lemmas

As our proof and example construction involves iteratively optimizing the subsidy scheme for the single-period problem instance, we begin by introducing the following definitions for clarity.

**Definition A.1.** *Given tolerance $\epsilon$, the problem instance $I = ((r_P(a_i), r_A(a_i))_i)$ is a single-period problem where $H = 1$, the agent is a globally $\epsilon$-IC agent, and $r_P, r_A$ are the principal and agent reward functions for actions $a_1, \cdots, a_{|A|}$. Given $\epsilon$ and problem instance $I$:*

- $V_P^{\Delta r}(I)$ *denotes the **principal value** under subsidy scheme $\Delta r$.*

- $V_A^{\Delta r}(I)$ *denotes the **maximal agent value** under subsidy scheme $\Delta r$.*

- $V_P^*(I)$ *denotes the **optimal principal value** under the optimal subsidy scheme $\Delta r^*$.*

- $V_A^*(I)$ *denotes the **maximal agent value** under the optimal subsidy scheme $\Delta r^*$.*

Based on the above definitions, we establish several useful properties of the subsidy scheme in the following lemmas, which will be employed in the proof of NP-hardness. Intuitively, the first lemma describes how the principal's value is determined when the agent adversarially reallocates probabilities in response to a given reward transfer. The second lemma characterizes the optimal reward transfer and the corresponding principal and agent values in a simple two-action instance. The third lemma analyzes how the optimal principal value in a three-action instance relates to the optimal values of its two-action sub-instances, providing useful bounds for iterative constructions.

**Lemma A.2.** *Let $I = ((a, b), (c, d))$ with $a > c$ and $b \geq d$. Under the subsidy scheme $\Delta r = 0$, the principal's final value is*

$$V_P^{\Delta r = 0}(I) = \frac{ab - a\epsilon - ad + c\epsilon}{b - d}.$$

*Proof.* To adversarially minimize the principal value under the constraint of global $\epsilon$-IC, it's obvious that the agent will choose to mix the first and the second action so that $V_A^{\Delta r = 0}(I) = b - \epsilon$. By

denoting the probability weight on the first action as $p$, we have

$$pb + (1-p)d = b - \epsilon \implies p = \frac{b - d - \epsilon}{b - d}.$$

Substituting into the principal's value formula,

$$V_P^{\Delta r = 0}(I) = pa + (1-p)c = \frac{ab - a\epsilon - ad + c\epsilon}{b - d}.$$

$\square$

**Lemma A.3.** *Let $I = ((a, 0), (0, 0))$ with $a > \epsilon$. Then the principal's optimal value is*

$$V_P^*(I) = (\sqrt{a} - \sqrt{\epsilon})^2,$$

*achieved by setting the reward transfer for the first action as $\sqrt{a\epsilon}$. Correspondingly, the agent's value under this optimal transfer is*

$$V_A^*(I) = \sqrt{a\epsilon}.$$

*Proof.* Clearly, the principal will allocate a positive subsidy to the first action only if doing so can yield a principal value exceeding zero. Consider a reward transfer $x \geq \epsilon$ assigned to the first action under a subsidy scheme $\Delta r$. For a given $x$, the principal's value is

$$V_P^{\Delta r}(I) = a - x - \frac{a\epsilon}{x}.$$

Maximizing over $x \geq \epsilon$, we obtain the optimal principal value

$$V_P^*(I) = \max_{x \geq \epsilon} V_P^{\Delta r}(I) = a + \epsilon - 2\sqrt{a\epsilon} = (\sqrt{a} - \sqrt{\epsilon})^2.$$

Consequently, the agent's value under the optimal agent value is

$$V_A^*(I) = \sqrt{a\epsilon}.$$

$\square$

**Lemma A.4.** *Let $I = (A_1, A_2, A_3)$ with $A_i = (r_P(a_i), r_A(a_i))$. Suppose for $i \in \{2, 3\}$, $r_P(a_1) + r_A(a_1) > r_P(a_i) + r_A(a_i) + \epsilon$. Define $I' = (A_1, A_2)$ and $I'' = (A_1, A_3)$. Then*

$$V_P^*(I) \leq \min\{V_P^*(I'), V_P^*(I'')\}.$$

*Proof.* According to Proposition 4.2, for any instance, the optimal reward transfer assigns a nonzero reward only to the first action. Applying concave maximization to instance $I$, we recall that the objective function is

$$F(x) = x V_{\text{sw}}^*(\hat{s}, h = 0) - V_x^*(\hat{s}, h = 0) - \frac{x}{1-x}\epsilon,$$

where $\pi_x = \arg\max_\pi \{x V_{\text{sw}}^\pi(\hat{s}, h = 0) - V_P^{\pi, \Delta r = 0}(\hat{s}, h = 0)\}$, and its derivative is

$$F'(x) = V_{\text{sw}}^*(\hat{s}, h = 0) - V_{\text{sw}}^{\pi_x}(\hat{s}, h = 0) - \frac{\epsilon}{(1-x)^2}.$$

By observation, we have

$$F'(0) = V_{\text{sw}}^*(\hat{s}, h = 0) - V_{\text{sw}}^{\pi_0}(\hat{s}, h = 0) - \epsilon,$$

where $\pi_0$ minimizes the principal's value, and $F'(x) \to -\infty$ as $x \to 1$. Therefore, there are two cases:

- There exists $\pi_0$ that chooses $a_1$ in instance $I$. Substituting into the derivative, we obtain $F'(0) < 0$, and the optimal solution is $x^* = 0$, which implies that the optimal subsidy scheme is $\Delta r^* = 0$. Similarly, the optimal schemes for $I'$ and $I''$ are also $\Delta r^* = 0$. Since action $a_1$ yields the highest social welfare, it also provides the largest agent value. Consequently, in all three instances, the agent will deterministically select action $a_1$, resulting in $V_P^*(I) = V_P^*(I') = V_P^*(I'')$.

- $\pi_0$ chooses an action other than $a_1$. In this case, based on the condition that for $i \in \{2, 3\}$, $r_P(a_1) + r_A(a_1) > r_P(a_i) + r_A(a_i) + \epsilon$, we have $F'(0) > 0$ and $x^* \in (0, 1)$ for instances $I$, $I'$ and $I''$. According to Proposition 4.2, the optimal subsidy assigns a positive transfer only to action $a_1$, and there exists an agent's adversarial action policy that mixes action $a_1$ with other actions in all three instances. Suppose the optimal subsidy schemes for instances $I$, $I'$, $I''$ are $\Delta r^*(I)$, $\Delta r^*(I')$, and $\Delta r^*(I'')$, respectively. Then we have

$$V_P^*(I) = \min\{V_P^{\Delta r^*(I)}(A_1, A_2), V_P^{\Delta r^*(I)}(A_1, A_3)\},$$
$$V_P^*(I') = V_P^{\Delta r^*(I')}(A_1, A_2),$$
$$V_P^*(I'') = V_P^{\Delta r^*(I'')}(A_1, A_3).$$

By the definition of the optimal subsidy scheme, we then obtain

$$V_P^{\Delta r^*(I)}(A_1, A_2) \leq V_P^{\Delta r^*(I')}(A_1, A_2), V_P^{\Delta r^*(I)}(A_1, A_3) \leq V_P^{\Delta r^*(I'')}(A_1, A_3),$$

which completes the proof.

$\square$

### A.7.2 SUPPLEMENTARY CALCULATIONS FOR EXAMPLE IN SECTION 5

In this section, we provide supplementary details regarding the computation of the optimal subsidy scheme and the corresponding principal value for both Markovian and non-Markovian agents. For the Markovian agent, there exists an optimal subsidy scheme $\Delta r^*$ in which the principal exclusively subsidizes actions $a_1$ and $a_2$. This is because, for any scheme that subsidizes actions other than $a_1$ and $a_2$, eliminating these additional subsidies increases the principal's value. To illustrate this, we consider the action governing the transition from $s_1$ to $s_3$ as an example, as the arguments for other actions are straightforward. When the subsidy on this action is reduced to zero, the resulting occupancy measure of the agent's policy increases for action $a_1$ and decreases for state $s_3$ and action $a_2$. Consequently, as long as the principal derives greater value from action $a_1$, this modified scheme yields a higher overall principal value. As we will demonstrate later, the optimal subsidy on action $a_1$ is at most 3, which substantiates this claim.

Let $x$ and $y$ denote the subsidies for actions $a_1$ and $a_2$, respectively, and let $p$ represent the probability of choosing action $a_2$ in state $s_2$. We begin by establishing preliminary bounds for the optimal subsidies, $x^*$ and $y^*$. First, we observe that $y^* \leq 0.21$. If $y > 0.21$, the principal's reward becomes negative; thus, the principal can always increase its value by reducing the subsidy on $a_2$. Second, it is straightforward to verify that the combination $(x = 1, y = 0, p = 1)$ is feasible and yields a principal value of $0.71$. Consequently, we must have $x^* \leq 3$: if $x \geq 3$, the optimal principal value is bounded by $0.21 + (3.21 - 3) = 0.42$, which is strictly less than $0.71$. Finally, the optimal subsidies must satisfy $x^* \geq y^*$: For any combination where $x < y$, the principal's value strictly decreases compared to the zero-subsidy baseline.

Then, by applying Lemma A.2, the principal's reward function with respect to $x$, $y$, and $p$ is derived as $f(x, y, p)$:

$$f(x, y, p) = \frac{1}{2}\left[3.21 - x + \frac{x - y - 3}{2 + x - (1 + y)p} + p(0.21 - y)\right].$$

Based on all above derivations, the optimization problem for the principal's value is defined as

$$\max_{y \in [0, 0.21]} \max_{x \in [y, 3]} \min_{p \in [y/(y+1), 1]} f(x, y, p)$$

where the lower bound on $y$ arises from the $\epsilon$-IC constraint. We first analyze the best response of $p$ given $x$ and $y$. The second-order derivative of $f$ with respect to $p$ is given by:

$$\frac{\partial^2 f}{\partial p^2} = \frac{(1 + y)^2(x - y - 3)}{(2 + x - (1 + y)p)^3}$$

Given the domains of $x$ and $y$, the function is concave with respect to $p$ when $p \geq 0$. Consequently, the optimal $p^*(x, y)$ lies at the boundaries: either $y/(1 + y)$ or $1$. This allows us to apply the

Envelope Theorem to analyze the derivatives with respect to $x$:

$$
\left.\frac{\partial f}{\partial x}\right|_{p=\frac{y}{1+y}} = \frac{1}{2}\left(-1 + \frac{5}{(2+x-y)^2}\right)
$$

$$
\left.\frac{\partial^2 f}{\partial x^2}\right|_{p=\frac{y}{1+y}} = -\frac{5}{(2+x-y)^3}
$$

$$
\left.\frac{\partial f}{\partial x}\right|_{p=1} = \frac{1}{2}\left(-1 + \frac{4}{(1+x-y)^2}\right)
$$

$$
\left.\frac{\partial^2 f}{\partial x^2}\right|_{p=1} = -\frac{4}{(1+x-y)^3}
$$

Based on the ranges of $x$ and $y$, the function is concave with respect to $x$. The stationary point is either $x = 1 + y$ or $x = \sqrt{5} - 2 + y$. We can thus further constrain the range of $x$ to $[y, 1+y]$ and investigate the first-order derivative of $f$ with respect to $p$ at $p = 0$:

$$
F(x, y) \triangleq \left.\frac{\partial f}{\partial p}\right|_{p=0} = \frac{(1+y)(x-y-3)}{(2+x)^2} + (0.21 - y)
$$

Examining the derivative with respect to $x$, we observe that $F(x, y)$ is monotonically increasing in $x$ for $x \in [0, 1+y]$ and $y \in [0, 0.21]$:

$$
\frac{\partial F}{\partial x} = \frac{(1+y)(-x+2y+8)}{(2+x)^3} > 0
$$

Let $h(y) \triangleq F(1+y, y)$; we now show that $h$ is monotonically decreasing with respect to $y$:

$$
h(y) = 0.21 - y - \frac{2(1+y)}{(3+y)^2}
$$

$$
h'(y) = -1 - \frac{2(1-y)}{(3+y)^3} < 0
$$

Thus, $h(y) \leq h(0) = 0.21 - 2/9 < 0$. Since $f$ is concave with respect to $p$, it follows that $f$ is monotonically decreasing with respect to $p$ in range $[0, 1]$. This demonstrates that the agent will deterministically choose $a_2$ at state $s_2$ when $x \in [ys, 1+y]$ and $y \in [0, 0.21]$. Therefore, for a given $y$, the optimal choice of $x$ is $1 + y$. Substituting $p = 1$ and $x = 1 + y$ into $f(x, y, p)$, the objective reduces to the following maximization problem:

$$
\max_{y \in [0, 0.21]} 0.71 - y
$$

Clearly, the optimal solution is $y^* = 0$, yielding a final principal value of $0.71$. Accordingly, we have $x^* = 1$ and $p^* = 1$.

For the non-Markovian case, we directly apply Lemma A.3 to determine that the optimal subsidy on action $a_2$ in state $s_3^2$ is $0.1$. Regarding the $s_1$ branch, we apply a method similar to the Markovian setting, using a modified principal value function:

$$
f(x, y, p) = \frac{1}{2}\left[3.21 - x + \frac{x - y - 3}{2 + x - (1+y)p}\right].
$$

Obviously, $p^* = 1$ since $y > 0$. By computing the derivatives with respect to $x$ and $y$, we find $x^* = 1$ and $y^* = 0$. Thus, the final optimal principal value is $0.605 + 0.01 = 0.615$.

### A.7.3 PROOF OF THEOREM 5.1

We prove the hardness by a reduction from the Maximum Independent Set problem.

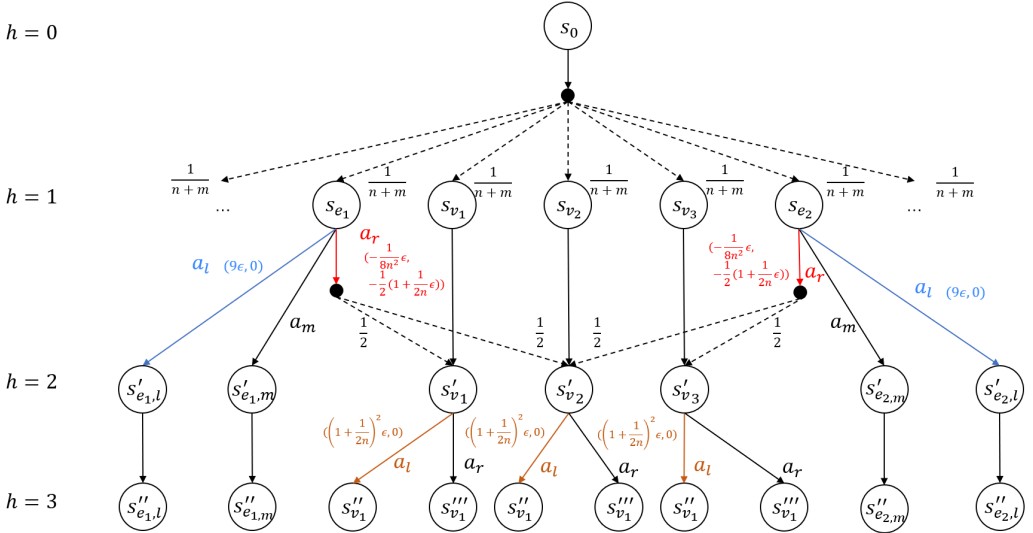

Figure 5: Illustrative Construction of the NP-Hardness Reduction Instance

**Construction**    Given a graph $G = \langle V, E \rangle$, we construct a corresponding problem instance as illustrated in Figure 5. Let $n = |V|$ and $m = |E|$, and assume $n \geq 2$.

Throughout the proof, we refer to an action with $r_P = 0$ and $r_A = 0$ as a *blank action*. The time horizon is set to $H = 4$, with a global initial state $\hat{s}$ that has only one blank action, which transitions uniformly to the vertex states $s_{v_1}, s_{v_2}, \ldots, s_{v_n}$ and the edge states $s_{e_1}, s_{e_2}, \ldots, s_{e_m}$.

For each vertex state $s_v$, there is a blank action leading to an intermediate state $s'_v$ for padding. The subsidy scheme at this intermediate state encodes whether the corresponding vertex $v$ is included in the maximum independent set. Two actions, $a_l$ and $a_r$, are available from $s'_v$ with the following specifications:

- $a_l$: $r_P(s'_v, a_l, h = 2) = \left(1 + \frac{1}{2n}\right)^2 \epsilon$, $r_A(s'_v, a_l, h = 2) = 0$, deterministically transitioning to state $s''_v$.
- $a_r$: a blank action that deterministically transitions to state $s'''_v$.

For each edge $e \in E$ connecting vertices $v_i$ and $v_j$, the corresponding edge state $s_e$ has three actions $a_l$, $a_m$, and $a_r$ designed to enforce the independent set constraints:

- $a_l$: $r_P(s_e, a_l, h = 1) = 9\epsilon$, $r_A(s_e, a_l, h = 1) = 0$, deterministically transitioning to state $s'_{e,l}$, which is then followed by a padding state $s''_{e,l}$.
- $a_m$: a blank action that deterministically transitions to state $s'_{e,m}$, which is then followed by a padding state $s''_{e,m}$.
- $a_r$: $r_P(s_e, a_r, h = 1) = -\frac{1}{8n^2}\epsilon$, $r_A(s_e, a_r, h = 1) = -\frac{1}{2}\left(1 + \frac{1}{2n}\right)\epsilon$, which transitions with equal probability to states $s'_{v_i}$ and $s'_{v_j}$.

Under this MDP construction, we claim that there exists an independent set of size $k$ in $G$ if and only if there exists a subsidy scheme $\Delta r$ in the MDP that allows the principal to achieve a reward of

$$\frac{\frac{k}{4n^2} + 4m}{n + m} \epsilon.$$

**If direction**    Given a size-$k$ independent set $V^* \subset V$ in graph $G$, we construct a subsidy scheme $\Delta r$ that achieves a principal value of $\frac{\frac{k}{4n^2}+4m}{n+m}\epsilon$. The scheme is defined as follows:

- For each $v \in V^*$, set $\Delta r(s'_v, a_l, h = 2) = 1 + \frac{1}{2n}\epsilon$ and $\Delta r(s'_v, a_r, h = 2) = 0$.

- For each $v \notin V^*$, set zero subsidy for both actions $a_l$ and $a_r$ at $(s'_v, h = 2)$.
- For each edge $e \in E$, set $\Delta r(s_e, a_l, h = 1) = 3\epsilon$ and leave actions $a_m$ and $a_r$ with zero subsidy.
- No subsidy is applied to any other actions.

With this subsidy scheme $\Delta r$, the agent greedily minimizes the principal value from bottom to top. First, consider a vertex state $s'_v$ and let $\pi$ denote the agent's action policy under $\Delta r$. By utilizing Lemma A.3, we obtain:

- For $v \in V^*$, $V_P^{\pi_{\Delta r}, \Delta r}(s'_v, h = 2) = \frac{1}{4n^2}\epsilon$ and $\overline{V}_A^{\Delta r}(s'_v, h = 2) = 1 + \frac{1}{2n}\epsilon$.
- For $v \notin V^*$, $V_P^{\pi_{\Delta r}, \Delta r}(s'_v, h = 2) = 0$ and $\overline{V}_A^{\Delta r}(s'_v, h = 2) = 0$.

Next, consider an edge state $s_e$, where $e$ connects vertices $v_1$ and $v_2$. There are two scenarios depending on whether one of the endpoints is in the independent set:

- If one endpoint is in $V^*$ (i.e., $v_1 \in V^*$ or $v_2 \in V^*$), the agent faces a single-period problem instance $((6\epsilon, 3\epsilon), (0, 0), (0, 0))$. Since two actions have identical principal and agent rewards, Lemma A.2 implies that the resulting principal value in $(s_e, h = 1)$ is $4\epsilon$.
- If neither endpoint is in $V^*$ (i.e., $V_A \notin V^*$ and $v_2 \notin V^*$), the agent faces the instance $((6\epsilon, 3\epsilon), (0, 0), (-\frac{1}{8n^2}\epsilon, -\frac{1}{2}(1 + \frac{1}{2n})\epsilon))$. Following the analysis in Lemma A.4, the agent chooses a mixture between $(a_1, a_2)$ or $(a_1, a_3)$. Applying Lemma A.2, we find with $n > 1$

$$V_P^{\Delta r=0}((6\epsilon, 3\epsilon), (0, 0)) - V_P^{\Delta r=0}\left((6\epsilon, 3\epsilon), \left(-\frac{1}{8n^2}\epsilon, -\frac{1}{2}\left(1 + \frac{1}{2n}\right)\epsilon\right)\right)$$

$$= 4\epsilon - \frac{15 + \frac{3}{2n} - \frac{1}{8n^2}}{\frac{7}{2} + \frac{1}{4n}}\epsilon$$

$$= \frac{(14 + \frac{1}{n}) - (15 + \frac{3}{2n} - \frac{1}{8n^2})}{\frac{7}{2} + \frac{1}{4n}}\epsilon$$

$$= \frac{-1 - \frac{1}{2n} + \frac{1}{8n^2}}{\frac{7}{2} + \frac{1}{4n}}\epsilon < 0,$$

which confirms that the final principal value remains $4\epsilon$.

Therefore, the total principal value under subsidy scheme $\Delta r$ is

$$V_P^{\pi_{\Delta r}, \Delta r}(\hat{s}, h = 0) = \frac{k \cdot \frac{1}{4n^2}\epsilon + (n - k) \cdot 0 + 4\epsilon \cdot m}{n + m} = \frac{\frac{k}{4n^2} + 4m}{n + m}\epsilon.$$

**Only if direction**  Suppose a subsidy scheme $\Delta r$ achieves $\frac{\frac{k}{4n^2} + 4m}{n + m}\epsilon$. for the principal. We show that this implies the existence of a size-$k$ independent set $V^* \subset V$ in $G$.

We first upper bound the maximum principal value achievable under any subsidy scheme $\Delta r$. There are two primary sources of principal value:

- Vertex states $s'_v$: by Lemma A.3, each vertex contributes at most $\frac{1}{4n^2}\epsilon$.
- Edge states $s_e$:  by Lemma A.3 and Lemma A.4, each edge contributes at most $V_P^*((9\epsilon, 0), (0, 0)) = 4\epsilon$.

Consequently, to attain the claimed principal value, at least $k$ vertex states must yield positive contributions. We claim that these vertices form an independent set. To see this, suppose otherwise: let $v_1$ and $v_2$ be connected by an edge $\bar{e}$. Since both $s'_{v_1}$ and $s'_{v_2}$ have nonzero principal values, the principal must provide a reward transfer of at least $\epsilon$ on action $a_l$ at both $s'_{v_1}$ and $s'_{v_2}$. Then, for action $a_r$ at $s_{\bar{e}}$, the agent's expected value is at least $(\frac{1}{2} - \frac{1}{4n})\epsilon$, while the principal's value is at most

$\frac{1}{8n^2}\epsilon$. We can upper bound the principal value from $(s_{\overline{e}}, h = 1)$ under any subsidy scheme $\Delta r$ as

$$V_P^{\pi_{\Delta r}, \Delta r}(s_{\overline{e}}, h = 1) \leq V_P^*\left((9\epsilon, 0), (\frac{1}{8n^2}\epsilon, (\frac{1}{2} - \frac{1}{4n})\epsilon)\right) \tag{A.18}$$

$$= \frac{1}{8n^2}\epsilon + V_0^*\left((9\epsilon - \frac{1}{8n^2}\epsilon, 0), (0, (\frac{1}{2} - \frac{1}{4n}))\epsilon)\right) \tag{A.19}$$

$$\leq \frac{1}{8n^2}\epsilon + V_0^*\left((9\epsilon, 0), (0, (\frac{1}{2} - \frac{1}{4n}))\epsilon)\right) \tag{A.20}$$

$$= \frac{1}{8n^2}\epsilon + V_0^*((\frac{17}{2} + \frac{1}{4n})\epsilon, 0), (0, 0)) \tag{A.21}$$

$$= \frac{1}{8n^2}\epsilon + \left(\sqrt{(\frac{17}{2} + \frac{1}{4n})\epsilon} - \sqrt{\epsilon}\right)^2$$

$$= \left(\frac{1}{8n^2} + (\frac{19}{2} + \frac{1}{4n}) - 2\sqrt{\frac{17}{2} + \frac{1}{4n}}\right)\epsilon$$

Inequality (A.18) follows directly from Lemma A.4 together with the observation that, in any single-period problem instance, simultaneously decreasing the principal's reward and increasing the agent's reward for an action can only reduce the optimal principal value. In equality (A.19), It is evident that subtracting the same value from the principal reward of each action and then summing afterwards does not affect the optimal solution. Inequality (A.20) arises from the fact that there is a pure principal reward increase in the first action. In equation (A.21), to obtain a strictly positive principal value, at least $(\frac{1}{2} - \frac{1}{4n})\epsilon$ must be subsidized on the first action. After such subsidy, as both actions now have the same agent reward, we set zero reward for both actions to sustain the relative value. Meanwhile, we can apply lemma A.3 to find the to optimal principal value.

Next, we upper bound the total principal reward across all sources:

- Principal reward from agent visiting $s_v$: at most $\frac{1}{4n^2}\epsilon$ per vertex, for at most $n$ vertices.
- Principal reward from agent visiting $s_e$ for edges $e \in E \setminus \{\overline{e}\}$: at most $4\epsilon$ per edge, for at most $m - 1$ edges.
- Principal reward from agent visiting $s_{\overline{e}}$: at most $\left(\frac{1}{8n^2} + (\frac{19}{2} + \frac{1}{4n}) - 2\sqrt{\frac{17}{2} + \frac{1}{4n}}\right)\epsilon$.

Summing over all contributions, the total principal value is

$$V_P^{\pi_{\Delta r}, \Delta r}(\hat{s}, h = 0)$$

$$\leq \frac{1}{n+m}\left[\left(\frac{1}{8n^2} + (\frac{19}{2} + \frac{1}{4n}) - 2\sqrt{\frac{17}{2} + \frac{1}{4n}}\right)\epsilon \right.$$

$$\left. + (m - 1) \cdot 4\epsilon + n \cdot \frac{\epsilon}{4n^2}\right].$$

Comparing with the claimed value, for $n \geq 2$ we obtain

$$\frac{\frac{k}{4n^2} + 4m}{n+m}\epsilon - V_P^{\pi_{\Delta r}, \Delta r}(\hat{s}, h = 0)$$

$$= \frac{\epsilon}{n+m}\left(\left(\frac{k}{4n^2} + 4\right) - \left(\frac{1}{8n^2} + \frac{19}{2} + \frac{1}{4n} - 2\sqrt{\frac{17}{2} + \frac{1}{4n}}\right) - \frac{1}{4n}\right) \tag{A.22}$$

$$\geq \frac{\epsilon}{n+m}\left(2\sqrt{\frac{17}{2}} - \frac{1}{8n^2} - \frac{11}{2} - \frac{1}{2n}\right) \tag{A.23}$$

$$\geq \frac{\epsilon}{n+m}\left(2\sqrt{\frac{17}{2}} - \frac{1}{32} - \frac{11}{2} - \frac{1}{4}\right) > 0$$

Here, inequality (A.22) follows from neglecting the $\frac{k}{4n^2}$ term, and inequality (A.23) is obtained by substituting $n = 2$. This contradiction demonstrates that any set of $k$ vertices yielding positive principal value must form an independent set, thereby completing the proof.

