# OpenReview forum: "Optimal Robust Subsidy Policies for Irrational Agent in Principal-Agent MDPs"
_ICLR.cc/2026/Conference — ICLR 2026 Poster_

### Official Review · Reviewer_v2YQ · 2025-10-28

**Soundness:** 2
**Presentation:** 2
**Contribution:** 2
**Rating:** 2
**Confidence:** 3

**Summary:**

This paper considers the Principal-Agent problem on Markov Decision Processes (MDP) where the Principal sets subsidies to the Agent in a form of an additive factor to the Agent's reward function. In turn, the Agent optimises its subsidised cumulative reward over a finite horizon MDP.
The main contributions of this paper are theoretical insights into optimal subsidy schemes under the assumption of bounded rationality of the Agent. In particular, three cases are considered; perfect rationality, $\epsilon$-incentive compatible Agent over episodes, and $\epsilon$-incentive compatible Agents over states.
In the first case, the authors show that the Principal's optimal payoff is the difference between the social welfare payoff and the Agent's unsubsidised payoff and provide an analytical solution to the optimal payoff scheme that only needs to subsidies actions corresponding to the social welfare maximizing actions.
In the second case, the authors show that the Principal's problem can be reformulated as a one-dimensional concave optimization, therefore, optimal subsidy schemes can be found with first-order optimization methods.
In the third case, the authors show that the Principal's optimization problem is NP-hard.

**Strengths:**

1. The paper is easy to follow and has a clear structure that builds from section to section.
2. Considering non-rational Agents in the Principal-Agent problem is a fundamental issue and having Principal actions that are robust to bounded rationality is crucial for applications.

**Weaknesses:**

1. The paper lacks discussion on how results connect to previous publications and what is the novelty of the contributions. While many recent publications are mentioned briefly in the Related Works section, without reading each of them it is hard to position the paper's contribution to the field. I strongly suggest the authors to provide more details for each Proposition and Theorem they prove what is the novelty compared to previous results.
2. The setting is limited to finite-horizon and finite state-action space MDPs with additive subsidy schemes. While this is a nice setup for toy experiments, for the results to be more relevant for the applications of Principal-Agent problems this is a clear limitation without any guidance on how the results might generalise to more complex spaces or Principal actions. Furthermore, as the authors also highlight the Conclusion this work is limited to settings where the Principal knows the Agent's reward function which does not hold for most applications, therefore, results when the Principal has to learn the Agent's incentives would be more relevant.
3. The paper lacks experimental results demonstrating the theoretical results. The Principal-Agent problem has many applications as the authors also highlight in the Introduction, therefore, empirical evaluations on some of the applications would solidify the theoretical results. Especially, ablations on the parameter $\epsilon$ to show how bounded rationality affect the outcomes of the problem.

**Questions:**

1. The authors claim the the Agent's optimal policy is deterministic in the perfectly rational case but might be stochastic in the bounded rational setting. Could they elaborate on this claim and at least provide pointers that show these results formally?
2. On page 7 Line 355-373, the authors discuss Markovian vs. Non-Markovian policies and claim that the Principal and the Agent adopt non-Markovian strategies in the globally $\epsilon$-IC case. Could the authors formalise the notion of non-Markovian in the case of finite-horizon MDPs and elaborate on what is the difference between non-Markovian and time-dependent?

---

> ### Author Response · Authors · 2025-11-18
>
> Thank you very much for reading our paper and providing thoughtful feedback. We deeply appreciate the time and effort you dedicated to reviewing our work.
>
> Regarding your first point about the related work, we apologize for not elaborating on it in sufficient detail due to space limitations. Overall, compared with prior work under the MDP framework, our paper proposes a **new model** that incorporates the principal’s subsidization cost into the objective function rather than as a constraint. Moreover, we provide a **new, clean and elegant closed-form solution** in the presence of a non-rational agent, specifically, a globally $\epsilon$-IC agent. As other reviewers noted, when dealing with boundedly rational agents, the principal–agent bilevel optimization problem is typically NP-hard. Thus, our main result is not only interesting but also provides a **analytical foundation** for more realistic scenarios.
>
> Regarding your second point, we acknowledge that our analysis assumes an idealized setting. However, **deriving a clear closed-form solution within this framework serves as a crucial benchmark for investigating more complex scenarios.** It allows us to precisely quantify the impact of different rationality levels on social welfare, providing a fundamental baseline against which practical problems can be analyzed.
>
> Regarding your third point, we agree that empirical evaluations of the relationship between $\epsilon$ and social welfare are important. In the appendix, we provide a detailed analytical characterization of the relationship between the optimal value $x^* $ and $\epsilon$. Furthermore, **Proposition 4.2 and 4.3** show how the agent’s optimal action policy and the resulting social welfare vary with $x^* $. We also provide a numerical example in **Figure 1** to illustrate this relationship.
>
> As for the two questions you raised, we apologize that the page limit prevented us from presenting all theoretical details, which may have led to misunderstandings regarding some specific concepts.
>
> ### Response to Question 1
>
> **Question: The authors claim that the Agent's optimal policy is deterministic in the perfectly rational case but might be stochastic in the bounded rational setting. Could they elaborate on this claim and at least provide pointers that show these results formally?**
>
> When the agent is **fully rational**, the agent’s problem reduces to maximizing its return in a standard MDP. It is well known that in such cases, the agent can achieve the maximum value using a deterministic policy.
>
> In contrast, when the agent follows a **globally $\epsilon$-IC requirement**, using stochastic action policies may further reduce the principal’s achievable payoff compared with deterministic action policies. For example, consider a post-subsidy MDP with a single state and two actions. The principal-agent reward for the first action is $(2, 2)$, and the principal-agent reward for the second action is $(0, 0)$. When $\epsilon=1$, if the agent is restricted to deterministic policies, it can only choose the first action, giving the principal a payoff of $2$. However, if the agent is allowed to randomize, it will mix equally between the two actions, reducing the principal’s payoff to $1$. This illustrates the fundamental difference between a fully rational agent and a globally $\epsilon$-IC agent, as the latter's feasible set may be constrained in ways that favor stochastic behavior.
>
>
>
> ### Response to Question 2
>
> **Question: On page 7 Line 355-373, the authors discuss Markovian vs. Non-Markovian policies and claim that the Principal and the Agent adopt non-Markovian strategies in the globally $\epsilon$-IC case. Could the authors formalise the notion of non-Markovian in the case of finite-horizon MDPs and elaborate on what is the difference between Non-Markovian and time-dependent?**
>
> In finite-horizon, **time-inhomogeneous** MDPs, both Markovian and non-Markovian policies can depend on time. The distinction lies in the dependence on the history:
>
> * A **Markovian policy** implies that, given the current state $s_t$ and the timestep $t$, the decision made does **not** depend on the past history (past states $s_{< t} $ and actions $a_{<t}$), i.e., $\pi: \mathcal{S} \times \mathcal{H} \to \Delta(A)$, where $\mathcal{H}$ denotes the set of time steps $\{0, 1, \ldots, H-1\}$.
>
> * A **Non-Markovian policy**, by contrast, conditions not only on the current state $s_t$ but also on the **entire history** of visited states and actions, i.e., $\pi: \mathcal{T} \to \Delta(A)$, where $\mathcal{T}$ denotes the set of all possible trajectories $\tau = (s_0, a_0, s_1, a_1, \ldots, s_t, a_t)$.
>
> We appreciate the opportunity to clarify some misunderstandings regarding definitions and welcome any further comments.

---

> > ### Comment · Reviewer_v2YQ · 2025-11-25
> >
> > Dear Authors,
> >
> > Thank you for addressing my concerns and questions.
> >
> > **W1**: I would strongly recommend to discuss in the Related Works section the differences in problem setting and assumptions of your work and the related publications (e.g. which papers consider finite or continuous spaces, highlight if your results strictly generalises or improves some other works). This would effectively highlight your contribution and make it clearer for the reader to assess your results.
> >
> > **W2**: I agree it is important to consider simplified settings to develop a foundation. I recommend at least discussing the effect of easing the assumptions to provide the reader some guidance how your results might generalise to broader problems. I can't find such discussion in the paper yet but please point me towards it if it is already included.
> >
> > **W3**: Thank you for amending the paper with Figure 1. Could you point me to the section discussing it and the setting it considers? I could not find any reference to Figure 1 in the main text. Minor point: I suggest adding x and y-axis labels to the plots and larger font size for better readability.
> >
> > **Q1**: My question referred to the fact that there exists a deterministic optimal policy but there might exists stochastic optimal policy as well. Do you assume that if a deterministic optimal policy exists the agent chooses that over any stochastic optimal policy? How would it affect the Principal if the agent would play an optimal but stochastic policy?
> >
> > **Q2**: Thank you for the clarification.

---

> ### Author Response · Authors · 2025-11-26
>
> Thank you again for the detailed comments following our initial rebuttal.
>
> ### W1
>
> To our best knowledge, we are unaware of any prior work that discusses how a principal can optimally subsidize an agent in an MDP that accounts for the agent's bounded rationality via the crucial $\epsilon$-IC constraint.
>
> As pointed out in our related work section and by other reviewers, solving most general principal-agent problems in MDP is computationally challenging (often proven to be NP-Hard)[1,2]. Our work provides a positive result by offering an analytical and tractable solution in such optimal robust subsidy problem.
>
> ### W2
>
> We acknowledge the request for extending our framework to more practical and general settings. Currently, we define the problem under the standard reinforcement learning objective, expressed in its expected form, and provide a clear and clean analytical solution. A natural and reasonable next step towards a more practical extension is the transition from linear objectives to convex objectives. In the context of reinforcement learning, Convex Reinforcement Learning (Convex RL) [3, 4, 5] is a direct and well-established generalization with applications in various domains(risk-averse RL[6], pure exploration[7], diverse skills discovery [8]).
>
> * In standard RL, the objective is linear in the occupancy measure $\mu$:
>   $\max_{\mu} \sum_{s,a,h} r(s,a,h)\mu(s,a,h)$
> * In Convex RL, the objective becomes a convex optimization problem over the occupancy measure $\mu$:
>   $\min_{\mu} f(\mu)$
>   where $f$ is a convex function.
>
> In future work, we can further extend the principal's and the agent's utility calculations to such a form. Our current analysis, which discusses fully rational and boundedly rational ($\epsilon$-IC) agents based on expected reward, provides a solid foundation for this generalization.
>
> ### W3
>
> In the example directly following Proposition 4.3 in Section 4.3, we discuss a single-step problem and illustrate how the optimal value $x^*$ and the social welfare gap $\delta_{\text{sw}}$change as $\epsilon$ varies. We cite Figure 1 for this illustration.
>
> The purpose of presenting this example is to demonstrate that the change in $\epsilon$ can lead to two different growth rates in the social welfare gap $\delta_{\text{sw}}$. We also provide a complementary example in Appendix A.2.2 where the social welfare gap just scales as $O(\sqrt{\epsilon})$. Together, these two examples highlight the non-trivial dependency on the bounded rationality parameter $\epsilon$. We also adds the x and y-axis labels for better readability.
>
> ### Q1
>
> Thank you for the clarification of your question. We confirm that we do not assume the agent must use a deterministic action policy throughout the paper. The conclusions we derive in our paper hold when the agent uses a stochastic action policy.
>
> To compare Section 3 (fully rational) and Section 4 (globally $\epsilon$-IC):
>
> * Fully Rational Case (Section 3): An optimal deterministic action policy for the agent always exists, but the agent is also free to choose any optimal stochastic action policy.
> * Boundedly Rational ($\epsilon$-IC) Case (Section 4): In the worst-case scenario for the principal, the agent's action policy may necessarily be stochastic to satisfy the $\epsilon$-IC constraint while minimizing the principal's utility.
>
> In summary, we emphasize that our framework and results do not impose a deterministic policy restriction on the agent in either setting.
>
> [1] Bollini, M., Bacchiocchi, F., Castiglioni, M., Marchesi, A., & Gatti, N. (2024). Contracting with a reinforcement learning agent by playing trick or treat.
>
> [2] Yang, K., & Zhang, H. (2024). Computational aspects of bayesian persuasion under approximate best response.
>
> [3] Hazan, E., Kakade, S., Singh, K., & Van Soest, A. (2019, May). Provably efficient maximum entropy exploration.
>
> [4] Zhang, J., Koppel, A., Bedi, A. S., Szepesvari, C., & Wang, M. (2020). Variational policy gradient method for reinforcement learning with general utilities.
>
> [5] Zahavy, T., O'Donoghue, B., Desjardins, G., & Singh, S. (2021). Reward is enough for convex mdps.
>
> [6] Hau, J. L., Petrik, M., & Ghavamzadeh, M. (2023). Entropic risk optimization in discounted MDPs.
>
> [7] Yarats, D., Fergus, R., Lazaric, A., & Pinto, L. (2021). Reinforcement learning with prototypical representations.
>
> [8] Liu, Hao, and Pieter Abbeel. Aps: Active pretraining with successor features.

---

### Official Review · Reviewer_v8gw · 2025-10-29

**Soundness:** 3
**Presentation:** 3
**Contribution:** 3
**Rating:** 6
**Confidence:** 4

**Summary:**

The paper studies a principal-agent problem. The agent performs actions in a Markovian environment. The principal offers subsidies to change the agent's reward. The agent acts optimally (either exactly or approximately) based on the subsidized rewards. The goal of the principal is to find a subsidy scheme so that the agent's best response gives the highest possible value to the principal. The authors studied both the case where the agent responds in a perfectly optimal way and the case where it does so only approximately. For both cases, the paper characterizes optimal subsidy schemes. The paper also analyzed the relation between the social welfare gap and the agent's rationality bound. Moreover, the paper considers a stronger constraint which arises when the gap between the agent's maximum attainable value and the policy they use is bounded below an epsilon threshold at every time step. The authors argue that in this case, the agent's induced policy may be non-Markovian, which introduces additional complexity to the problem. To work around, they considered an alternative greedy state-wise epsilon-IC condition, but even with that the problem remains NP-hard.

**Strengths:**

I think the problem studied is interesting and natrual. The paper is very clear and easy to follow. I didn't check the full proofs but the proof sketches are helpful for understanding the ideas, and the arguments look sound. The characterization results about optimal subsidy schemes and their relation to social welfare maximizing policies are very interesting, so is the relation between the social welfare gap and epsilon. The paper is overall solid and I enjoy reading it.

**Weaknesses:**

- The part in Section 5 could be explored a bit more. The authors introduced an alternative type of agent in an attempt to overcome the complexity caused by history-dependency, but even that is intractable. The paper would be stronger if a tractable alternative is introduced.

- Some related works are missing, but this is minor. The idea of subsidizing the agent's reward is a special case of the reward design problem. There are many papers in this direction which also adopt a worst-case perspective in the face of uncertainty about the agent's response; see, e.g., Admissible Policy Teaching through Reward Design, Banihashem et al., 2022. Similar principal-agent subsidy problems have also been studied under fairness considerations: Envy-free Policy Teaching to Multiple Agents, Gan et al., 2022. And there are similar principal-agent models based on Markov games, e.g. Stochastic Principal-Agent Problems: Efficient Computation and Learning, 2023.

**Questions:**

- Does your proof of Theorem 4.1 also imply that in the fully rational case, it is also always optimal for the principal to use a Markovian subsidy scheme.

- Would you be able to show that the problem against a state-wise epsilon-IC agent (Definition 5.1) is NP-hard?

---

> ### Author Response · Authors · 2025-11-18
>
> Thank you very much for taking the time to read our paper and provide thoughtful feedback.
>
> Regarding the additional related work you mentioned, we are grateful for your broader literature suggestions and have incorporated the relevant references into the paper.
>
> Regarding the weaknesses you pointed out, we acknowledge that our current analysis of the state-wise setting represents a preliminary exploration into this domain. Our primary objective in presenting the negative results is to characterize the significant computational obstacles inherent in this setting. By progressively introducing Definitions 5.1 and 5.2, we demonstrate that even under relaxed constraints, the problem remains computationally intractable. We emphasize this difficulty to clarify the landscape, with **the hope that identifying these barriers will guide future work toward a definition that is both non-trivial and computationally feasible.**
>
> **We further recognize that solvable special cases do exist.** For instance, in an MDP with deterministic and action-invariant transitions (where the next state for any pair $(s,h)$ is identical across all actions), globally $\epsilon$-IC and state-wise $\epsilon$-IC become equivalent, rendering the problem straightforward. However, such cases are structurally degenerate and do not offer meaningful insights for general models.
>
>
> ## Response to Question 1
>
> **Question: Does your proof of Theorem 4.1 also imply that in the fully rational case, it is also always optimal for the principal to use a Markovian subsidy scheme.**
>
> Yes, your observation is correct.
>
>
>
> ## Response to Question 2
>
> **Question: Would you be able to show that the problem against a state-wise epsilon-IC agent (Definition 5.1) is NP-hard?**
>
> We thank the reviewer for raising this interesting point. While we have not yet derived a formal proof of NP-hardness for the problem under Definition 5.1, we conjecture that the problem is indeed NP-hard. Our reasoning relies on two key observations:
>
> 1. We have established that the agent can adopt a non-Markovian policy to minimize the principal's reward.
>
> 2. Crucially, this non-Markovian dependence is not limited to a short history window; a decision at timestep $h$ may depend on a state from the distant past. This can be demonstrated by a straightforward extension of the examples in Figure 2. For instance, we can insert arbitrary padding states between $s_1$ and $s_3$, and between $s_2$ and $s_3$, such that the optimal policy in $s_3$ remains critically dependent on the trajectory's initial state.
>
> Consequently, we believe that optimizing against a non-Markovian agent requires an exponential number of mappings to fully describe the non-Markovian action policy space. This strongly suggests that a polynomial-time solution is unlikely, thereby implying NP-hardness.

---

### Official Review · Reviewer_dbNJ · 2025-11-01

**Soundness:** 3
**Presentation:** 3
**Contribution:** 3
**Rating:** 6
**Confidence:** 4

**Summary:**

This paper studies a principal-agent Markov Decision Process (MDP) problem where the principal can offer state-action-dependent subsidies to influence the agent's action policy.  The paper considers both perfectly rational (best-responding) agent as well as $\epsilon$-best-responding agent.  The paper shows that:

1. For perfectly rational agents, the principal's optimal subsidy can be characterized easily: it is the highest achievable social welfare minus the agent's default value.
2. For $\epsilon$-best-responding agents, taking the worst case for the principal among the $\epsilon$-best-responding policy sets, the paper shows that the principal's maximin problem can be converted into a single-dimensional concave maximization problem that admits an efficient solution.
3. Lastly, the authors consider a special case of $\epsilon$-best-responding -- state-wise $\epsilon$-IC -- in which case the optimal subsidy turns out to be NP-hard to compute.

**Strengths:**

(S1) The theoretical result for $\epsilon$-best-responding agent, where the authors show that the maximin problem can be converted into a single-dimensional concave problem (using a duality approach) is very interesting.  The fact that such an optimization problem can be solved efficiently is not obvious at all, especially given some previous works on $\epsilon$-best-response in other principal-agent problems, such as information design [Yang & Zhang, NeurIPS 2024, Computational Aspects of Bayesian Persuasion under Approximate Best Response].

(S2) Good theoretical coverage.  The main $\epsilon$-best-response model is in the ex-ante sense: the agent computes the overall expected utility.  The authors also consider the state-wise $\epsilon$-best-response model, establishing a clear boundary between tractable and intractable cases.  And both Markovian and non-Markovian policies are discussed.

(S3) The paper is very well written, with rigorous definitions, helpful examples, and intuitive proof sketches.

**Weaknesses:**

(W1) A related literature is "steering agents in games by payments", e.g., [1, 2, 3], which studies how a principal can add payment (subsidy) to a game to induce certain outcomes among the players.  This literature studies multi-player games, which are more general than the single-agent problem in this paper.  But on the other hand, this literature usually considers static or repeated game, which is less general than this paper's MDP model.  [2, 3] consider no-regret learning agent, which is a form of $\epsilon$-best-responding agent considered by this paper.  Given the conceptual similarities, I think a discussion to this literature is needed.  Nevertheless, I view this as a minor weakness because the techniques are different due to modeling differences.



---------------------------------

[1] Monderer & Tennenholtz, 2004, K-Implementation.

[2] Zhang et al, EC 2024, Steering No-Regret Learners to a Desired Equilibrium.

[3] Zhang et al, 2025, Learning a Game by Paying the Agents.

**Questions:**

## Questions for the authors

(Q1) Do you have any results for **Markovian** state-wise $\epsilon$-IC agent?  Namely, if the agent uses a Markovian policy that is $\epsilon$-optimal at every state, can the principal find the optimal subsidy efficiently?  (I am not sure whether Theorem 5.1 applies to Markovian state-wise $\epsilon$-IC.)



## Suggestions

* For the future direction "consider scenarios in which the principal does not have prior knowledge of the agent's reward function or the value of $\epsilon$, such as in a learning-based setting", [3] is very related.

---

> ### Author Response · Authors · 2025-11-18
>
> Thank you very much for taking the time to read our paper and for providing such valuable feedback. We agree that the literature you cited is highly relevant to our work, and we have incorporated the key insights into our revised manuscript, which are highlighted in blue.
>
> ## Response to Question
>
> **Question: Do you have any results for Markovian state-wise $\epsilon $-IC agent? Namely, if the agent uses a Markovian policy that is $\epsilon$-optimal at every state, can the principal find the optimal subsidy efficiently? (I am not sure whether Theorem 5.1 applies to Markovian state-wise $\epsilon $-IC.)**
>
> Although a formal proof remains to be established, we conjecture that the principal's problem of finding the optimal subsidy scheme for a Markovian state-wise $\epsilon $-IC agent is NP-hard. This conjecture stems from the belief that, even given a subsidy scheme, **the agent's subproblem of finding its own optimal action policy is highly likely to be NP-hard.**
>
> This conjecture is informed by an analysis of the feasible set of action policies:
>
> * If the action policy is represented by a state-wise policy $\pi$: The optimization objective and constraints involve terms similar to $E_\pi[\sum_{t=h}^{H-1}r_A(s_t, a_t, h_t)]$. Since these expressions involve continuous products of multiple $\pi$ terms, the feasible set for $\pi$ is non-convex.
> *  If the action policy is constructed using occupancy measures $\mu$: The additional state-wise constraints imposed by the state-wise $\epsilon$-IC agent, compared to a globally $\epsilon$-IC agent, still result in a non-convex feasible set. (see the example below)
>
> Consequently, without a more refined characterization of the agent's best response, deriving a definitive complexity proof for the principal's overall problem remains challenging.
>
> ### Example for Non-Convex Feasible Set of Occupancy Measure
>
> Consider the following MDP with horizon $H=3$. For simplicity, we introduce an "ending state" ($s_{end}$) where, upon entry, the agent can only execute a zero-reward action until the maximum timestep.
>
> * The MDP has 4 non-ending states.
> * The initial state is $s_1$.
> * $s_1$: Actions $a_1$ and $a_2$ lead deterministically to $s_2$ and $s_3$, respectively.
> * $s_2$: Actions $a_3, a_4, a_5$. $a_4$ leads to $s_4$. $a_3$ and $a_5$ lead to $s_{end}$.
> * $s_3$: Only action $a_6$, which leads to $s_4$.
> * $s_4$: Actions $a_7$ and $a_8$.
>
> Let $r$ and $\mu$ be vectors where $r_i$ and $\mu_i$ represent the reward and occupancy measure for action $a_i$, respectively.
>
> The core of our construction uses $a_5$ combined with $\epsilon$ to set an upper bound on the agent's achievable reward, which is then used to construct a non-convex feasible set for the occupancy measure at state $s_2$.
>
> The state-wise IC constraint at state $s_2$ (at time $t=1$), when using an occupancy measure $\mathbf{\mu}$, can be written as:
>
> $$V_A^{\mu}(s_2, 1) = \frac{1}{\mu_3 + \mu_4 + \mu_5}\cdot(\mu_3r_3 + \mu_4r_4 + \mu_5r_5) + \frac{\mu_4}{\mu_3 + \mu_4 + \mu_5}\cdot\frac{1}{\mu_7 + \mu_8} \cdot(\mu_7r_7 + \mu_8r_8) \geq r_5 - \epsilon$$
>
>
>
> **Observation**: The left-hand side of the inequality contains quadratic terms, such as $\mu_4\mu_7$ , which intuitively suggests a non-convex feasible set.
>
> **A Concrete Construction**:
> Assume the agent's reward is $r_A = (0, 0, 5, 15, 15.1, 0, 1, 10)$ and set $\epsilon=1$. Clearly, the maximum value achievable from $s_2$ is $r_5 = 15.1$. The constraint requires $V_A^{\mu}(s_2, 1) \geq 15.1 - 1 = 14.1$.
>
> Consider two feasible occupancy measures:
>
> *  $\mu^{(1)} = (0.5, 0.5, 0.45, 0.05, 0, 0, 0.1, 0.9)$
> *  $\mu^{(2)} = (0.5, 0.5, 0.05, 0.45, 0, 0, 0.9, 0.1)$
>
> Substituting these into the constraint yields:
>
> * $V_A^{\mu^{(1)}}(s_2, 1) = 14.19 > 14.1$ (Feasible)
> * $V_A^{\mu^{(2)}}(s_2, 1) = 14.19 > 14.1$ (Feasible)
>
> Now consider their convex combination: $\mu^{(3)} = 0.5\mu^{(1)} + 0.5\mu^{(2)}$.
>
> * $V_A^{\mu^{(3)}}(s_2, 1) = 12.75 < 14.1$ (**Infeasible**)
>
> Since the convex combination of two feasible points is infeasible, the feasible set of the occupancy measure is non-convex.

---

> > ### Comment · Reviewer_dbNJ · 2025-11-28
> >
> > When I tried to understand the authors' response regarding Markovian policies, I got confused by some parts of the paper.
> >
> > (1) First, regarding notations: Section 2 explicitly says that the agent uses a Markovian policy $\pi: S\times H \to \Delta(A)$, and defines the value functions $V^\pi(s, h)$ and $Q^\pi(s, a, h)$ accordingly.  I think these notations are specific to Markovian policies, because $V^\pi(s, h)$ only takes the current state and time index into account, not including history.  But the authors start to talk about non-Markovian (history-dependent strategies) after Theorem 4.1 and in Section 5.  How should I think about the notations $V^\pi(s, h)$ for non-Markovian policies?
> >
> > (2) Since the proof of Theorem 4.1 (globally $\epsilon$-IC case) uses the $V^\pi(s, h)$ notations, is this theorem specific to Markovian strategies?
> >
> > (3) In the end of Section 4.1, the "second observation" paragraph says that "a non-Markovian scheme can be transformed into a Markovian one by augmenting the state space to encode the relevant history".  Although this is true, it will change the structure and solution of the MDP, and also enlarge the state space.  If the number of relevant histories is exponential, then this is an exponential increase in the size of the state space of the MDP, which will cause computational difficulty. Is that true?  Or is it the case that the number of relevant histories in your setting is just polynomial?  (For example, the principal might use a grim-trigger strategy: once the agent deviates from the principal's wish, the principal will start to punish the agent (by giving no subsidy) in the future forever.  In this case, we just need to know whether the agent has deviated in the past, which reduces the number of relevant histories to polynomial, I guess.)
> >
> > (4) In Section 5, figure 2, why is the non-Markovian policy where "at $s_3^1$, the agent always selects $a_2$, while at $s_3^2$, the agent mixes between two acgtions with equal probability" a state-wise $\epsilon$-IC strategy for the agent?  Shouldn't the agent also selects $a_2$ at $s_3^2$?  Also, related to (1), the definition of state-wise $\epsilon$-IC uses the Markovian notation $V^\pi(s, h)$, so I don't know how to interpret it in the non-Markovian setting.
> >
> > (5) This is a suggestion: Given the simplicity of Markovian strategy, and its popularity in the MDP literature, I think it is OK if this paper only focuses on Markovian strategies.  Discussing the non-Markovian strategies but using the Markovian notation causes confusion.

---

> ### Author Response · Authors · 2025-11-28
>
> Thank you for bringing up the concern regarding non-Markovianity. Initially, for the sake of presentation and accessibility, we intentionally limited our notation to the Markovian setting, on which we based the derivation of Theorem 3.1 and Theorem 4.1. Crucially, **the content and proofs of these theorems demonstrate that the optimality results are consistent between non-Markovian and Markovian policies**. Therefore, it is sufficient for us to analyze and compute the solution within the simplified Markovian framework.
>
> Specifically, a non-Markovian MDP can be expanded into an equivalent tree-structured Markovian MDP by duplicating states to encode history information. Let $\mathcal{C}(s)$ denote the set of all replicated states corresponding to an original state $s$, distinguished by their unique histories. Consequently, a non-Markovian policy in the original formulation maps directly to a standard Markovian policy in this expanded MDP. To extend our notation, we replace instances of the pair $(s, h)$ with the replicated counterpart $(\overline{s}, h)$, where $\overline{s} \in \mathcal{C}(s)$. This transformation applies analogously to the value functions $V$ and $Q$.
>
> Based on the conclusions of Theorems 3.1 and 4.1, when considering the expanded Markovian MDP, we find that all replicated states share the same optimal $\Delta r^{*}$. This key finding confirms the fundamental consistency of the optimal solution between the non-Markovian and Markovian settings.
>
> Taking Theorem 4.1 as an example: if we expand the MDP and assume that the optimal subsidy scheme itself is also non-Markovian, we can directly apply the MDP result. For any replicated state $\overline{s} \in \mathcal{C}(s)$ corresponding to the same original state $s$, we have:
>
> $$\Delta r^{* }(\overline{s}, a, h) = V_{x^{* }}^{* }(\overline{s}, h) - Q_{x^{* }}^{* }(\overline{s}, a, h)$$
>
> Evidently, the optimal subsidy scheme is identical across all these replicated states, confirming that the optimal Markovian subsidy scheme yields the exact same results as any optimal non-Markovian scheme. Furthermore, since the optimal solution is identical, we only need to solve for the Markovian optimal subsidy schemes, thus eliminating the need for a specialized non-Markovian solver and avoiding potential issues related to exponential time complexity.
>
> Regarding Example 5.1, our main objective is to demonstrate a scenario where the agent, by adopting a non-Markovian policy, successfully reduces the overall value obtained by the principal. Specifically, when the agent uses a non-Markovian policy, the original notation can be interpreted as:
>
> $$V_A^{\pi}(s,h) = \sum_{\overline{s}\in \mathcal{C}(s)} \text{Pr}(\overline{s}_h = \overline{s} | s_h=s) V_A^\pi(\overline{s}, h)$$
>
> $$\overline{V}_A (s,h) = \underset{\overline{s} \in \mathcal{C}(s) }{\max} \overline{V}_A(\overline{s},h) = \overline{V}_A(s,h)$$
>
> where $\text{Pr}(\overline{s}_h = \overline{s} | s_h = s)$ is the probability the agent arrives at $\overline{s}$ in timestep $h$ conditioned on the agent arrives at $s$ at timestep $h$. The agent's behavior presented in the example strictly adheres to this definition and illustrates the disparity that can arise between the non-Markovian and Markovian agent policy spaces. Specifically, for the non-Markovian policy $\pi$, which chooses $a_2$ at $s_3^1$ and equally mixes two actions at $s_3^2$,  the resulting agent value is $V_A^\pi(s_3, 2) = 1/2 \cdot 2 + 1/2 \cdot(1/2 \cdot 2 +1/2\cdot 0) = 1.5$,  while the maximum potential value is $\overline{V}_A^\pi(s_3, 2) = 2$. It achieves a principal reward of $1.5$. In contrast, the principal's reward under any Markovian policy is at least $2$.

---

### Official Review · Reviewer_vNuN · 2025-11-05

**Soundness:** 3
**Presentation:** 2
**Contribution:** 2
**Rating:** 2
**Confidence:** 4

**Summary:**

This paper considers the principal-agent MDP problem where the agent is assumed to be boundedly rational in the sense that the agent may take any suboptimal responses up to epsilon gap. This problem is modeled as a robust, minimax game: the principal tries to find a subsidy policy that maximizes social welfare, while the agent responds with the worst-possible, sub-optimal deviation. The paper shows that, if the agent response is globally epsilon suboptimal, the optimal subsidy scheme can be effectively determined. Meanwhile, if the agent response is the state-wise ϵ-IC, finding the optimal subsidy is NP-hard.

**Strengths:**

This paper makes a good effort to extend the setting of principal-agent MDP problems. Modeling agent’s bounded rationality moves beyond the perfect-world assumptions of classical game theory and makes the work directly applicable to real-world systems where users are not perfect optimizers.

**Weaknesses:**

It is unclear why the paper chose to focus on the welfare maximization problem instead of the more common objective to maximize the principal’s utility. I suspect the results of this paper will no longer hold under the objective to maximize the principal’s utility. Therefore, the authors should clarify on this modeling choice and explain how the current solution might rely on the welfare maximization objective.

**Questions:**

What's the motivation or is there any technical result that this work focus on the welfare maximization objective?

---

> ### Author Response · Authors · 2025-11-18
>
> We thank the reviewer for taking the time to assess our paper and for providing such valuable feedback.
>
> We apologize if our initial presentation caused any misunderstanding regarding the principal’s objective. To clarify, the central assumption in our paper is that the **principal’s sole objective is to determine an optimal subsidy scheme that maximizes its own final revenue**. The principal is not assumed to be responsible for, nor to directly optimize, social welfare.
>
> In particular, Theorems 3.1 and 4.1 characterize this optimal, revenue-maximizing subsidy scheme for the principal under different agent assumptions.
>
> The subsequent analysis of social welfare investigates the impact of $\epsilon $ (bounded rationality) on the system. We observed that, in the fully rational case, the principal’s optimal revenue-maximizing strategy coincides with the strategy that maximizes social welfare. Our analysis explores how this alignment changes as bounded rationality increases.
>
> We appreciate the opportunity to clarify this point and welcome any further comment.

---

### Meta-Review · Area_Chair_3Wez · 2025-12-28

**Summary:**

The paper investigates a principal-agent problem within Markov Decision Processes (MDPs) where a principal provides subsidies to influence the actions of a bounded rational agent. It establishes that while optimal subsidies for perfectly rational agents align with social welfare maximization, the problem for "globally $\epsilon$-incentive-compatible" agents reduces to a tractable one-dimensional concave optimization.

The expected score is quite boarderline. After reading the paper myself, I lean towards acceptance as the paper established that the optimal robust subsidy scheme problem simplifies to a one-dimensional concave optimization, which is an interesting observation and tool for future studies.

**Reviewer Concerns:**

Optimization objective raised by Reviewer vNuN is addressed.
Literature discussion raised by Reviewer dbNJ is addressed.

**Reviewer Scores:**

Reviewer vNuN's score is based on a misunderstanding of the paper and is excluded in the decision making.
Reviewer dbNJ and Reviewer v8gw are likely to maintain the score 6.
Reviewer v2YQ is likely to increase the score to 4 due to partially addressing their concerns but do not provide any experiments.

---

### Decision · Program_Chairs · 2026-01-26

Accept (Poster)